# Sea ice–air interactions amplify multidecadal variability in the North Atlantic and Arctic region

Jiechun Deng 🔟 [1]✉ & Aiguo Dai 🔟 [2]✉

Winter surface air temperature (Tas) over the Barents–Kara Seas (BKS) and other Arctic regions has experienced rapid warming since the late 1990s that has been linked to the concurring cooling over Eurasia, and these multidecadal trends are attributed partly to internal variability. However, how such variability is generated is unclear. Through analyses of observations and model simulations, we show that sea ice–air two-way interactions amplify multidecadal variability in sea-ice cover, sea surface temperatures (SST) and Tas from the North Atlantic to BKS, and the Atlantic Meridional Overturning Circulation (AMOC) mainly through variations in surface fluxes. When sea ice is fixed in flux calculations, multidecadal variations are reduced substantially (by 20–50%) not only in Arctic Tas, but also in North Atlantic SST and AMOC. The results suggest that sea ice–air interactions are crucial for multidecadal climate variability in both the Arctic and North Atlantic, similar to air-sea interactions for tropical climate.

[1] Key Laboratory of Meteorological Disaster, Ministry of Education (KLME)/Joint International Research Laboratory of Climate and Environmental Change (ILCEC)/Collaborative Innovation Center on Forecast and Evaluation of Meteorological Disasters (CIC-FEMD), Nanjing University of Information Science and Technology, Nanjing 210044, China. [2] Department of Atmospheric and Environmental Sciences, University at Albany, State University of New York, Albany, NY 12222, USA. ✉email: jcdeng@nuist.edu.cn; adai@albany.edu

Superimposed on a rapid warming trend, Arctic surface air temperature (Tas) since the early 20th century also exhibits large multidecadal variations[1–3] that cannot be explained by concurring monotonic increases in atmospheric greenhouse gases (GHGs). Model simulations[1,4,5] show that internal variability can generate similar low-frequency variations in Arctic Tas; and multidecadal variations in poleward energy transport associated with the Atlantic Multidecadal Oscillation (AMO) and Atlantic Meridional Overturning Circulation (AMOC)[6] have been identified as a leading cause of the Arctic Tas multidecadal variations[4,7,8]. It is suggested[4,8] that above-normal oceanic heat transport from the North Atlantic into the Barents–Kara Seas (BKS) and other Arctic regions reduces sea-ice cover (SIC) there, which allows the Arctic Ocean to absorb more energy during the summer but release more heat to the air to cause warmer Tas during the winter. On the other hand, Arctic sea-ice loss[9,10] is found to weaken the AMOC in coupled model simulations[10–12], and sea ice in the subpolar Atlantic and Nordic Seas might play a role in Atlantic multidecadal variability (AMV)[13,14]. These findings raise an important question: Can the multidecadal variations in AMV (or AMO) and AMOC, a major source of multidecadal climate variability around the Atlantic and beyond[6], and thus the oceanic heat transport itself be influenced by sea-ice changes and variations? Furthermore, it is unclear whether poleward energy transport alone can directly cause large multidecadal variations in Arctic Tas without the participation of sea ice. Answers to these questions have major implications for the formation mechanisms of the AMOC and AMV[14] and for our upcoming climate, as models project large sea-ice loss over the BKS and other Arctic regions in the coming decades to centuries[15–17].

Here we analyze observations and various model simulations, which exhibit realistic patterns for SIC and Tas multidecadal variability (cf. Figs. 1a, b, 2a, and Supplementary Fig. 1), to show that winter Tas multidecadal variations largely disappear in the Arctic and subpolar North Atlantic when sea ice variations no longer exist, either by fixing it in surface flux calculations (see Methods) or after sea ice melts away under GHG-induced warming. Furthermore, the associated AMV, AMOC and SIC variations are also substantially reduced under these conditions, which implies a strong amplification of the multidecadal variability in the North Atlantic and Arctic region by sea ice–air interactions. Results for annual-mean and other seasons are similar with smaller magnitudes (see Methods). Thus, our new findings suggest that Arctic and subpolar North Atlantic sea ice–air interactions are crucial for Arctic and North Atlantic multidecadal variability, similar to air-sea interactions for tropical climate variability. The poleward energy transport alone cannot cause large multidecadal variations in Artic Tas without the sea ice–air two-way interactions, and that such Tas variations over the BKS and other Arctic regions will likely weaken if the current sea–ice margins continue to retreat. Such a change might also affect Eurasian weather and climate, as warm BKS Tas anomalies are associated with more persistent atmospheric blocking over the Ural Mountains that can cause cold anomalies over central Eurasia through enhanced cold advection[18,19]. Furthermore, the projected sea-ice loss will also weaken and eventually eliminate sea ice–air interactions in the subpolar North Atlantic, which may contribute to the weakening of the AMOC and AMV and their variability that have major impacts on European and global climate[6,20].

## Results

### Multidecadal climate variability and the role of sea ice–air coupling over the Arctic and North Atlantic. Observations show larger multidecadal variations in SIC and Tas along Arctic sea-ice

margins, such as the BKS, Greenland-Norwegian Seas (GNS), and Labrador Sea-Davis Strait (LSDS) (Fig.1a, b). These variations are anti-correlated over time (with correlation coefficients $r$ ranging from $-0.76$ to $-0.92$, $p < 0.05$) and both are related to AMV, the multidecadal variations in North Atlantic SST (NASST) (Fig. 1c), with either little time lag (e.g., for LSDS) or some delay (e.g., for GNS and BKS) (Fig. 1d–f). However, the AMV-associated NASST anomalies (up to ~0.2 °C) are much smaller than the Tas anomalies over the three Arctic regions, and by the time the NASST anomalies being advected to the Arctic regions through ocean currents, their magnitudes should be even smaller as the exchange of heat with the air would damp the SST anomalies along the way. This suggests that the direct heating of the air over the three Arctic regions by the AMV-associated SST anomalies is small and cannot explain the large multidecadal Tas anomalies over the Arctic regions, which are up to ~2 °C over the LSDS and ~1.5 °C over the BKS (Fig. 1d, f).

These observed multidecadal variations are reproduced approximately in our CESM1 pre-industrial control run (CTL) or 1% per year $CO_2$ increase run (1%$CO_2$), with large multidecadal variations for both SIC and Tas over the Arctic regions and for NASST (i.e., AMV) (Fig. 2a, d and Supplementary Fig. 1a, d). However, when the sea ice–air two-way interactions are cut off in our pre-industrial control run (CTL_FixedIce) or 1%$CO_2$ run (1%$CO_2$_FixedIce), which only allow the atmosphere and oceans to affect sea ice but not the other way under a fixed sea-ice cover (with a seasonal cycle) for flux calculations only (see Methods), such multidecadal variations in Tas (Fig. 2c and Supplementary Fig. 1c) and SIC (Supplementary Fig. S2a–f) over the Arctic regions weaken substantially along the sea-ice margins, especially over the GNS and BKS where large SIC variations (and thus strong sea ice–air coupling) are seen in the CTL or 1%$CO_2$ runs. We also found that multidecadal SST variations are reduced moderately over the northern North Atlantic and Nordic Seas in both FixedIce runs (Fig. 2d–f and Supplementary Fig. 1d–f). For example, Tas multidecadal variability over the LSDS and AMV weakens by about 36% and 31% (relative to CTL), respectively, in CTL_FixedIce, although SIC's multidecadal variability weakens only slightly over the LSDS (Fig. 3a, b). Similar variability reductions for both SIC and Tas are also seen in the BKS region in CTL_FixedIce (by ~49% for Tas and ~16% for SIC) compared to CTL (Supplementary Fig. 3a, b), and such reductions are even larger (by ~70% for Tas and ~19% for SIC) under increasing $CO_2$ before year ~150 (i.e., 1% $CO_2$_FixedIce relative to 1%$CO_2$) (Supplementary Fig. 4).

The large multidecadal temperature variations induced by the sea ice–air interactions can cause apparent warming or cooling trends over decadal-multidecadal periods over many Arctic regions, such as BKS, where a large decadal warming trend was observed from 1997 to 2009 (~2.69 °C/decade; Fig. 1f and Supplementary Fig. 5a). Such a decadal trend is also seen in our CTL run over certain decadal periods of similar length (e.g., the top five percentiles show a mean trend of ~2.99 °C/decade; Supplementary Fig. 5b). However, these strongest BKS decadal warming trends are reduced substantially (by ~55%) to ~1.36 °C/ decade when the sea ice–air interactions are cut off in CTL_FixedIce (Supplementary Fig. 5c). As a result, the recent decadal warming trend over the Arctic or BKS region seen in ERA5 is much less likely to occur without the sea ice–air coupling than the case with the coupling, with the occurrence probability of a similar or larger trend for the Arctic-mean and BKS trends increased, respectively, from 2.14% to 5.78% and 0.00% to 3.64% from CTL_FixedIce to CTL (Supplementary Fig. 5d, e). This implies that the recent rapid warming in the Arctic or BKS region can arise from the multidecadal variability amplified by the sea ice–air interactions (although it is a small-probability event), which also contributes to the concurring winter cooling over

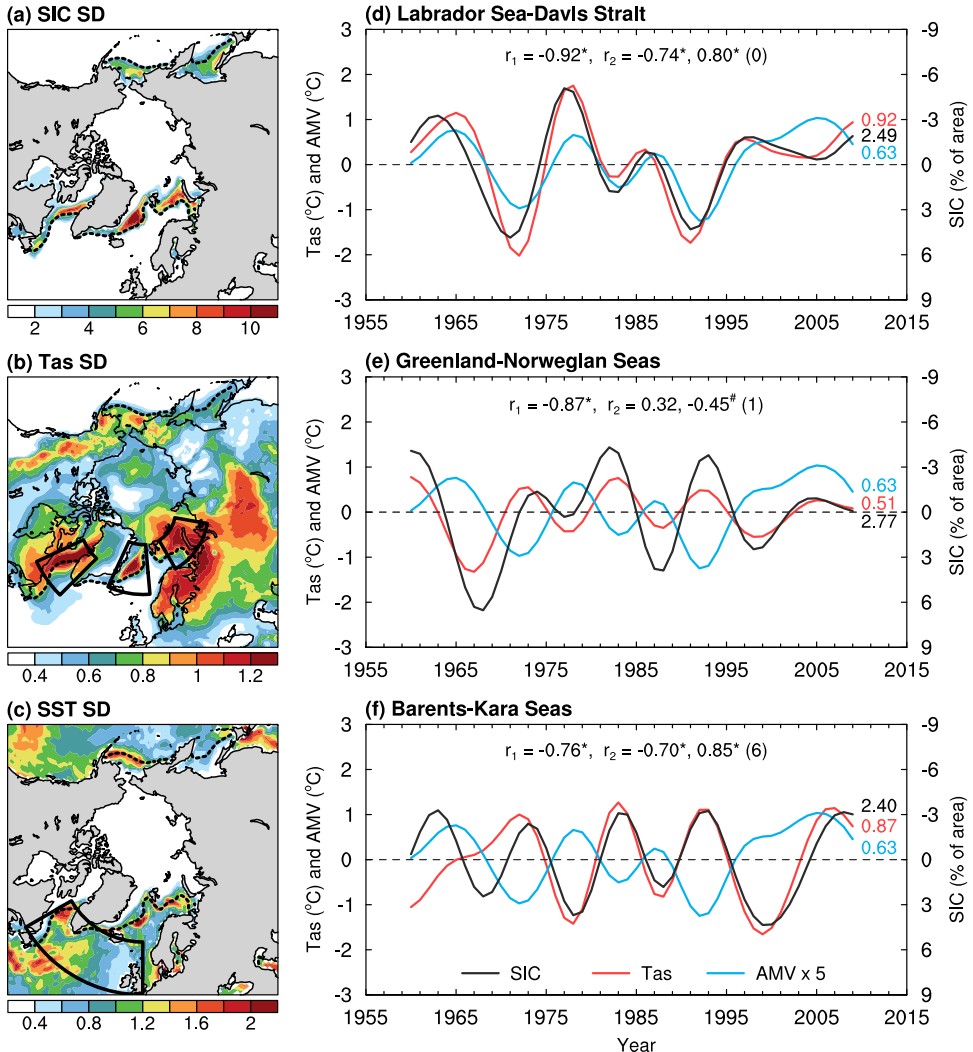

**Fig. 1 Multidecadal variability in the North Atlantic and Arctic region from ERA5. a–c** Distributions of the standard deviation (SD) of the 10–90-year band-pass filtered DJF-mean anomalies (with the forced signal removed; see Methods) in (**a**) sea-ice cover (SIC, in % of area), (**b**) surface air temperature (Tas, in °C), and (**c**) sea surface temperatures (SST, in °C; multiplied by 5) north of 50°N during 1950–2019 from ERA5 reanalysis. The outlined areas in (**b**) define, from west to east, the Labrador Sea and Davis Strait (LSDS; 45°–65°W, 55°–70°N), the Greenland-Norwegian Seas (GNS; 25°W–5°E, 65°–80°N), and the Barents–Kara Seas (BKS; 30°–80°E, 70°–80°N); and the black dashed contour in (**a–c**) denotes the climatological DJF-mean sea-ice edge (for sea-ice concentration = 10%). **d–f** Anomaly time series of the filtered DJF-mean SIC (black, in % of area; right y-axis, increases downward) and Tas (red, in °C; left y-axis; over oceans only within each region) averaged over the (**d**) LSDS, (**e**) GNS, and (**f**) BKS with the forced signal removed. The blue curve denotes the Atlantic multidecadal variability (AMV) index (in °C; multiplied by five to use the same left y-axis), defined as the similarly filtered DJF-mean SST anomalies averaged over the northern North Atlantic (0°–60°W, 50°–65°N; outlined in (**c**)) using ERA5. The SD of each curve is given in the respective color on each panel. The correlation coefficient $r_1$ is between SIC and Tas, and $r_2$ are, from left to right, the peak correlation coefficients with AMV leading SIC and Tas by the years in the parentheses. The superscript "*" ("#") indicates the correlation is statistically significant at the 5% (10%) level based on a resampling technique (see Methods).

Eurasia via atmospheric cold advection[21]. However, the relative roles of internal variability and external forcing in causing the recent Arctic warming trends still require further investigations.

These results suggest that the sea ice–air two-way interactions act to amplify not only the multidecadal Tas anomalies (and thus multidecadal Tas trends) over the Arctic regions, but also Arctic SIC variations and AMV. While we expect the Tas anomalies to weaken in CTL_FixedIce and 1%CO$_2$_FixedIce given the large amplification effect of sea-ice loss on surface warming shown previously[8,22], it is surprising that both SIC and NASST (i.e., AMV) variations also weaken substantially when fixed SIC is used in calculating surface fluxes over the Arctic and subpolar North Atlantic (note that SIC itself is not fixed but allowed to evolve dynamically in these model runs).

As sea ice melts away over the BKS, GNS and LSDS after year ~150 in the 1%CO$_2$ run, the sea ice–air interactions would weaken or even disappear completely over these regions. As a result, multidecadal SIC (Supplementary Fig. 2g–i) and Tas (Supplementary Fig. 6a) variations would weaken greatly over these regions. For example, multidecadal SIC and Tas anomalies over the BKS diminish gradually during years ~50–190 with declining sea ice and become close to zero thereafter when sea ice (and thus its interactions with the atmosphere) is mostly gone over this region (Supplementary Fig. 4a). On the other hand, as little sea ice exists in the northern North Atlantic after year ~150 in the 1%CO$_2$ run, multidecadal NASST variability would also weaken due to the absence of sea ice–air interactions by then (Supplementary Fig. 6b). We notice that the ice margins move

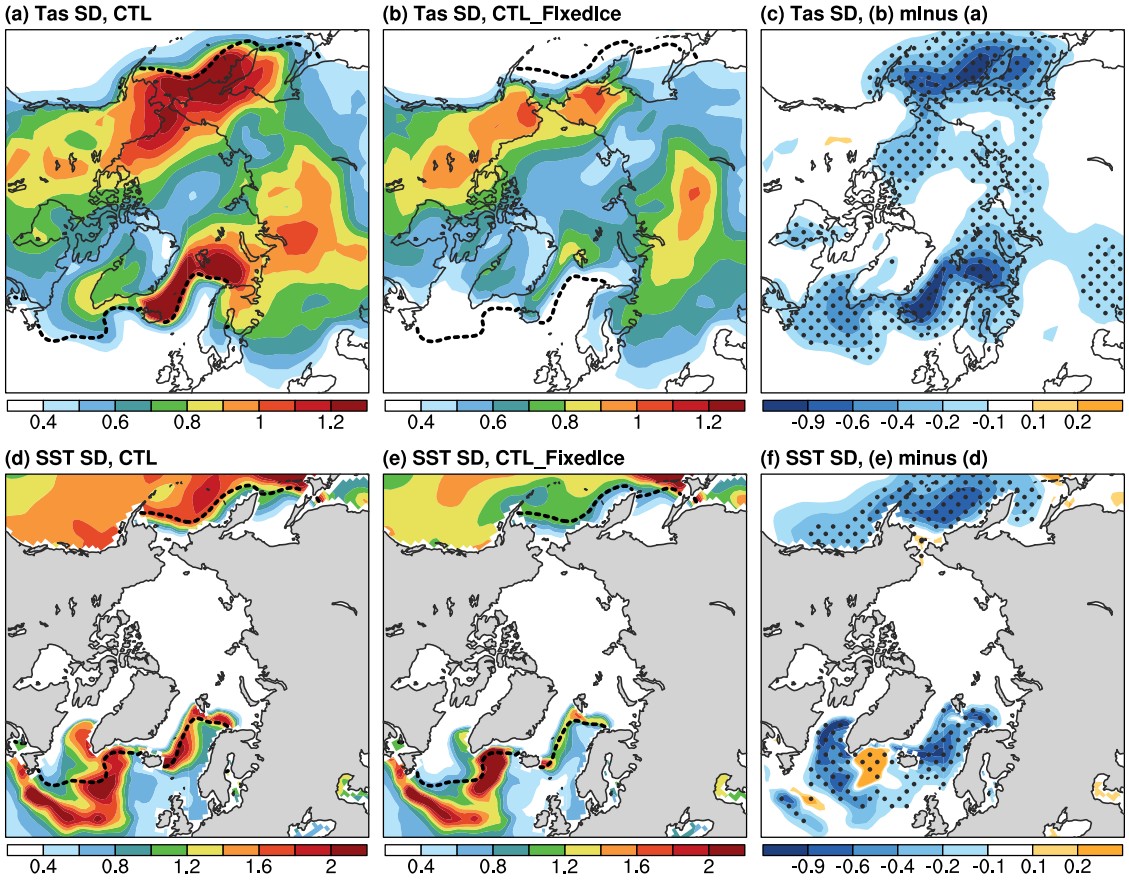

**Fig. 2 CESM1-simulated multidecadal temperature variability and its sea ice-induced difference. a–c** Distributions of the standard deviation (SD) of the 10–90-year band-pass filtered DJF-mean Tas anomalies (in °C) north of 50°N from the CESM1 (**a**) CTL and (**b**) CTL_FixedIce runs and (**c**) their difference (i.e., CTL_FixedIce minus CTL) during years 11–490. **d–f** Same as (**a–c**) but for the similarly filtered DJF-mean SST anomalies (in °C; multiplied by five as in Fig. 1). The black dashed contour in (**a**) and (**d**) (**b**, **e**) denotes the climatological DJF-mean sea-ice edge (for sea-ice concentration = 10%) over the same period from CTL (CTL_FixedIce). The stippling in (**c**), (**f**) indicates that the SD difference is statistically significant at the 5% level based on a *F*-test.

poleward under increasing $CO_2$, together with their associated multidecadal SIC and Tas variability (albeit weakened). This result suggests that the locations and magnitudes of multidecadal variations in Arctic SIC and Tas and NASST greatly depend on the locations of ice margins where the sea ice–air coupling is strongest. In other words, if subpolar and Arctic sea ice continues to retreat or melt away, its ability to cause such large multidecadal variations will diminish.

Consistent with the CESM1 results, other climate models also simulate large multidecadal SIC and Tas variability over the GNS, BKS, and LSDS along the ice margins and NASST variability during the historical period (1920–2019), and these multidecadal variations would weaken greatly or almost disappear in the 23rd century when winter SIC is mostly gone over the subpolar North Atlantic, BKS and other Arctic regions (Supplementary Figs. 2j–l and 6). Although the 23rd century climate includes many other changes, the results are at the least qualitatively consistent with our CESM1 results. These results suggest that the sea ice–air two-way interactions not only play a crucial role for Arctic Tas multidecadal variations, but also increase the low-frequency variability in Arctic SIC and NASST.

**Sea ice-induced positive feedback loop through surface flux changes.** The AMV-associated relatively small SST anomalies can be advected from the North Atlantic to Arctic ice margin zones in 2–6 years by upper-ocean currents in both observations and our

CTL run (Supplementary Fig. 3d), and the sea ice–air coupling allows sea ice to respond to and amplify these ocean-induced SST anomalies. We further found that such multidecadal SIC anomalies are anti-correlated with anomalies in winter surface upward longwave (LW) radiation ($r = -0.56$, $p < 0.05$) and turbulent heat fluxes ($r = -0.64$, $p < 0.05$) (Supplementary Fig. 7a); that is, large upward energy flux anomalies are collocated with large multidecadal sea-ice decline, especially over the GNS, BKS, and LSDS regions (Supplementary Fig. 7c). This is because a multidecadal sea-ice decline induced by a positive SST anomaly allows the warm Arctic Ocean to release large amounts of heat and LW radiation to warm up the frigid winter Arctic air greatly, as shown previously[8,22]. The resultant warmer air would in turn increase downward LW radiation (ref. [23]. and Supplementary Fig. 8a) and further melt sea ice, leading to a positive feedback loop that amplifies the SST-induced variations.

When fixed SIC is used in calculating all the surface fluxes, such multidecadal relationship between SIC and surface energy fluxes weakens (for turbulent fluxes with $r = -0.17$, $p < 0.1$) or even reverses (for upward LW radiation with $r = 0.7$, $p < 0.05$) (Supplementary Fig. 7b). This is because in our FixedIce runs, increased upward LW radiation (which is decoupled with internal SIC) would lead to surface cooling and thus more sea ice (Supplementary Fig. 7d), in contrast to the fully coupled run in which the internal SIC largely determines these fluxes so that low SIC leads to more open waters and increased oceanic heat release through surface fluxes in winter. In other words, the SIC change

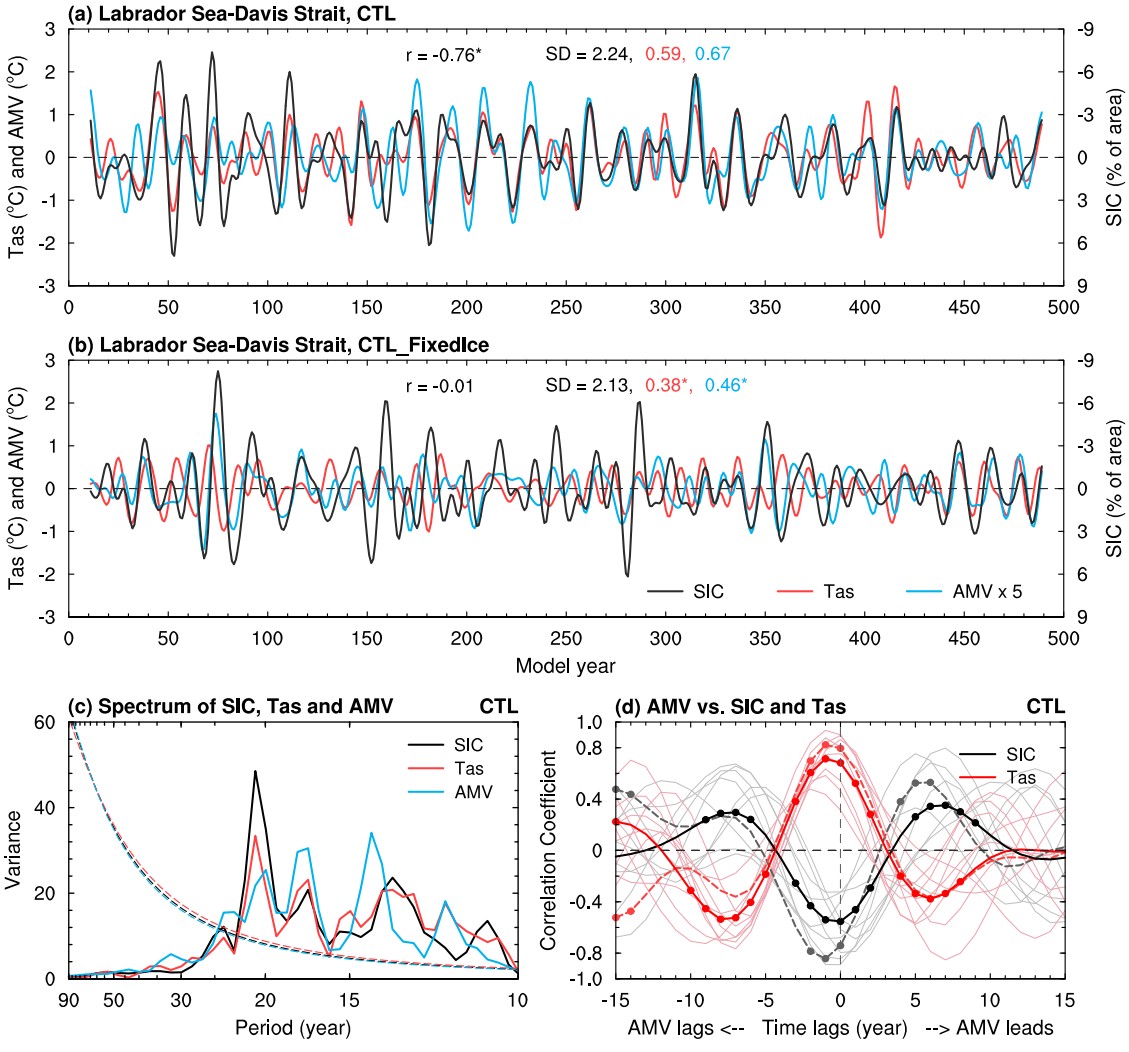

**Fig. 3 Relationships among AMV, regional sea ice and air temperature in CESM1.** Filtered time series of DJF-mean anomalies of SIC (black; in % of area; right *y*-axis, increases downward) and Tas (red; in °C, left *y*-axis) averaged over the Labrador Sea and Davis Strait defined in Fig. 1b and the AMV index (blue; in °C, multiplied by five in order to use the same left *y*-axis) from the CESM1 (**a**) CTL and (**b**) CTL_FixedIce runs from years 11–490. A 10 to 90-year Lanczos band-pass filter was used. The correlation coefficient (*r*) is between SIC and Tas. The SD of each curve is given in the respective color on (**a**, **b**). The superscript "*" and "#" indicate the correlation (the SD difference) is statistically significant at the 5% and 10% levels, respectively, based on a resampling technique (a *F*-test) (see Methods). **c** Power spectrum (standardized to use the same *y*-axis) of the time series shown in (**a**). The dashed curves are for the 95% confidence bound. **d** Lead-lag correlation coefficients of the AMV with SIC (black) and Tas (red) over the LSDS from ERA5 reanalysis as shown in Fig. 1d (dashed lines) and the CESM1 CTL run as shown in (**a**) (thick solid lines). The thin solid lines with the respective colors are for nine 50-year (similar to the length of the filtered ERA5 data) segments from CTL, and their ensemble mean are similar to the respective thick solid lines (which are for all years). The dots indicate the correlation coefficient is statistically significant at the 5% level based on a resampling technique (see Methods).

is a cause of the change in upward LW radiation in the fully coupled runs, but a result in the FixedIce runs. Without the sea ice-induced amplification effect when the sea ice–air interactions are cut off, multidecadal surface flux anomalies are reduced (Supplementary Fig. 7b), leading to weak multidecadal Tas variations in our CTL_FixedIce run (Figs. 2b, 3b, and Supplementary Fig. 3b).

Surprisingly, multidecadal variations in winter SST from the northern North Atlantic to the BKS also weaken significantly when Arctic sea ice is fixed in flux calculations (Fig. 2f) or melts away in the 23rd century (Supplementary Fig. 6b). While we expect the lack of large Tas variations over the BKS and GNS to weaken the positive feedback effect on SST anomalies over these two regions through changes in downward LW radiation (Supplementary Fig. 8b), we are surprised to see that the absence of sea ice–air coupling also weakens the multidecadal SST

variability from the Labrador Sea to the subpolar North Atlantic east of it and the Nordic Seas (Fig. 2f and Supplementary Fig. 6b). We emphasize that in the positive feedback loop, the lower tropospheric temperature and humidity will be altered by the surface upward fluxes, leading to changes in downward LW radiation, thus providing an atmospheric influence on sea ice and surface temperature[24]. In the fully coupled CTL run, a consistent power peak for an oscillation around 21 years was found in AMV (which includes SSTs over the northern North Atlantic) and both SIC and Tas over the LSDS (Fig. 3c), and their relationships are almost simultaneous with little time lag (Fig. 3d). This suggests that the multidecadal SST variability over the Labrador Sea and other subpolar Atlantic regions is closely coupled to the sea ice–air interactions, mainly through the SIC-induced anomalies in surface latent heat flux (LHF) (Supplementary Fig. 9) and other surface energy fluxes. When a constant SIC is used in calculating

surface fluxes in the CTL_FixedIce run, such SIC-associated multidecadal LHF variability decreases by about 74% over the LSDS (compared to CTL; see below), thereby weakening SST variations over this region.

During the 23rd century when subpolar sea ice melts away, large decreases in multidecadal LHF variability are also found over the northern North Atlantic and the Nordic Seas (Supplementary Fig. 10), which are collocated with large reductions in NASST's variability (Supplementary Fig. 6b). This suggests that, without the SIC-induced large LHF variations through the sea ice–air coupling, the multidecadal NASST variations would become much weaker. Again, although the future changes in LHF (and thus NASST) variations under such a high emission scenario can be caused by many other factors, these results are qualitatively consistent with our CESM1 results. Thus, the surface-flux induced amplification and sea ice–air feedback mentioned above can amplify the SIC and SST anomalies and cause large Tas changes from the North Atlantic to the Arctic through SIC's impact on surface fluxes.

**Multidecadal variability of AMOC.** Our CESM1 CTL run simulates large multidecadal AMOC variations centered at around 1.5 km depth between ~40°–50°N (Fig. 4a). It is surprising that such AMOC multidecadal variability weakens (by about 20%) over the northern North Atlantic (north of 40°N) when the sea ice–air interactions are cut off in our CTL_FixedIce run (Fig. 4b, c). In the fully coupled CTL run, the AMOC index is clearly correlated with SIC (and thus LHF) over the LSDS on multidecadal time scales centered around 21 years (Figs. 5a and 6a, b), with SIC ($r = \sim-0.4$, $p < 0.05$) and LHF ($r = \sim0.5$, $p < 0.05$) leading the AMOC index by 4–5 years (Fig. 6c). We found that multidecadal oscillations in upper-ocean salinity and density are also anti-correlated with SIC variations over the LSDS (Fig. 5a), implying a connection among multidecadal variability in SIC and other upper-ocean conditions. Composite analyses further show that the sea ice–air coupling in CTL allows multidecadal winter SIC decrease (increase) over the LSDS to cause large positive (negative) multidecadal anomalies in surface net water flux (i.e., evaporation minus precipitation or E–P) and LHF over exposed waters; and such enhanced (weakened) surface evaporation and heat loss from ocean to air would in turn increases (decreases) sea surface salinity (SSS), upper-ocean density and thus North Atlantic Deep Water (NADW) formation (measured by ocean mixed layer depth) over the LSDS region on multidecadal timescales (Fig. 7a, b). These variations in density and resultant NADW formation would lead to deeper and stronger (shallower and weaker) AMOC multidecadal anomalies in about 3–5 years during the high (low)-LHF anomaly periods associated with low (high) SIC anomalies (Figs. 6d and 7c).

However, when fixed SIC is used in calculating the surface fluxes, the SIC-associated multidecadal variability in E–P and LHF over the LSDS is reduced by about 27% and 74% ($p < 0.05$) relative to CTL, respectively, leading to considerably weakened variations in upper-ocean salinity and density (Fig. 5b, c) and thus ocean mixed layer depth in and around the LSDS (Supplementary Fig. 11a–c). Without the sea ice–air interactions, the SIC-associated LHF anomalies become weak in CTL_FixedIce, resulting in smaller multidecadal anomalies in SSS and upper-ocean density and thus NADW formation (Fig. 7d, e); that is, such SIC variations cannot enlarge multidecadal anomalies in upper-ocean salinity and density (and thus NADW formation) over the LSDS through its associated surface fluxes. As a result, the resultant AMOC variations become weak and shallow (Fig. 7f).

These multidecadal relationships are also found in the fully coupled piControl runs by other climate models, and reported previously with different lags[25]. In most of these piControl runs,

the AMOC index is also anti-correlated with SIC variations over the LSDS (Supplementary Fig. 12) but with a longer lag time (~10 years) (Fig. 6e). The AMOC index is also similarly correlated with a lag of ~3–5 years with the SIC-related multidecadal variations in E–P and LHF (Fig. 6e), and thus the SSS and ocean mixed layer depth (Fig. 6f), although the AMOC's connection with the multidecadal SIC variation itself exhibits substantial spread among the models (black curve in Fig. 6e) likely due to the weak and uncertain SIC-LHF relationship in these models (Supplementary Fig. 9). Thus, these piControl runs also show that higher (lower) LHF anomalies induced by multidecadal SIC decrease (increase) over the LSDS and its adjacent North Atlantic regions can lead to increased (decreased) SSS and thus NADW formation, thereby enhancing (weakening) AMOC's multidecadal anomalies (Supplementary Fig. 13), consistent with the results from the CESM1 CTL run.

In summary, the results from the CESM1 and other climate models suggest that SIC variations in the subpolar North Atlantic can amplify the multidecadal variability in AMOC and NASST through its impact on E–P, LHF and other surface fluxes, and that sea ice–air two-way interactions over the LSDS play a crucial role for large multidecadal variations in SIC, NADW formation and AMOC. When the Arctic sea ice–air coupling weakens or disappears, AMOC's multidecadal variability and its associated NASST variations would weaken.

We should note that the SIC-induced AMOC variations (i.e., lower SIC leading to stronger AMOC through increased LHF and other fluxes and vice versa) can also provide a positive feedback on SIC through AMOC-induced poleward heat transport. For example, a stronger (weaker) AMOC would transport more (less) heat into the LSDS and other subpolar North Atlantic, thereby further reducing (increasing) SIC there. This may have occurred in CTL_FixedIce for its mean SIC and AMOC strength. The CTL_FixedIce run shows increased mean SIC (Supplementary Fig. 14) and weakened mean AMOC (Fig. 4c) compared to CTL, together with reduced mean SST, LHF, SSS, MLD and ocean density over the subpolar North Atlantic (Supplementary Fig. 14). These mean changes from CTL to CTL_FixedIce are qualitatively consistent with the multidecadal anomalies associated with high SIC anomalies and weak AMOC discussed above; presumably they are produced by the same surface flux-based processes and the AMOC-induced positive feedback mentioned above. Note that the enhanced LHF can cool the surface, but the cooling is overcome by other warming processes, leading to mostly positive SST anomalies during years with large LHF in the CESM1 (Supplementary Fig. 15).

On the other hand, our standard $1\%CO_2$ run shows that the AMOC would become weaker and shallower under increasing $CO_2$ (contours in Fig. 4d–f) likely due to increased ocean stratification caused by surface warming and freshening in the North Atlantic, as shown previously[16,26,27] and possibly by sea-ice loss and associated Arctic amplification under large $CO_2$ increases[28]. Meanwhile, its multidecadal variability also weakens over the subpolar North Atlantic in future warmer climates (shading in Fig. 4d–f), as reported previously[29,30], which would weaken multidecadal anomalies in NASST[6]. Although we would expect such a variability reduction as AMOC's variability amplitude is closely related to AMOC's mean strength (Fig. 4d–f), the above surface flux-based processes could also weaken AMOC's multidecadal variations as sea ice and thus its interactions with the atmosphere decrease in future warmer climates. As multidecadal SIC variations would weaken greatly over the LSDS in the CESM1 $1\%CO_2$ run or in the 23rd century in the CMIP5/CMIP6 model simulations, the associated multidecadal LHF anomalies over this region would also become much weaker due to the lack of the sea ice–air interactions (Supplementary Figs. 16 and 17), leading to greatly reduced multidecadal variability in the associated SSS and

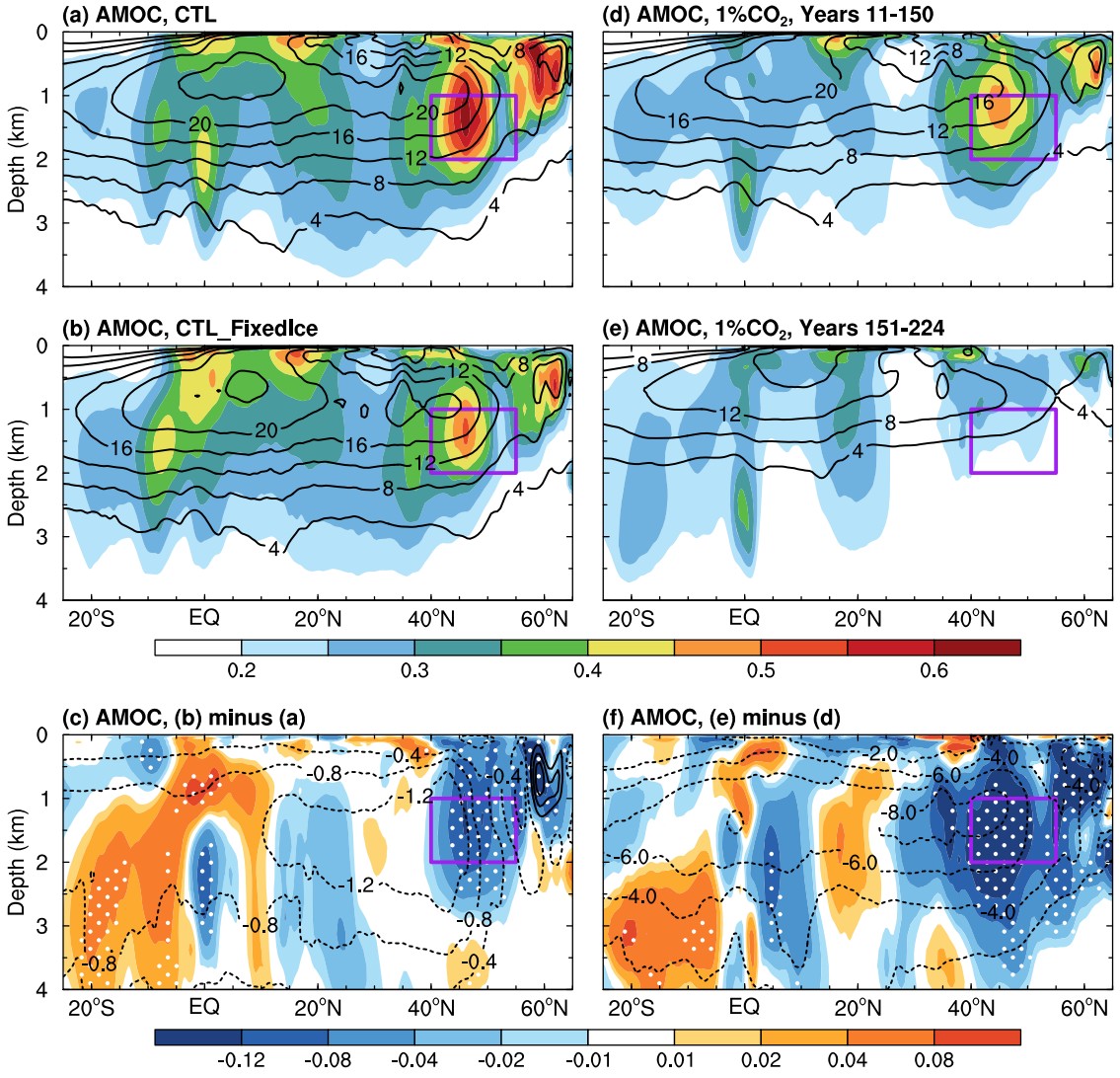

**Fig. 4 CESM1-simulated differences in Atlantic Meridional Overturning Circulation (AMOC) climatology and variability.** Distributions of the standard deviation (SD) of the 10 to 90-year band-pass filtered DJF-mean AMOC stream-function anomalies (shading; in Sv) from the CESM1 (**a**) CTL and (**b**) CTL_FixedIce runs and (**c**) their difference (i.e., CTL_FixedIce minus CTL) during years 11 to 490. The contours represent the climatological DJF-mean AMOC zonal-mean stream-function (in Sv) from (**a**) CTL and (**b**) CTL_FixedIce and (**c**) the CTL_FixedIce-minus-CTL difference during years 11–490, and the solid and dashed contours are for positive and negative values, respectively. The outlined area (40°–55°N, 1–2 km) is used to define the AMOC index in Fig. 5. (**d**–**f**) Same as (**a**–**c**), respectively, but from the CESM1 1%CO$_2$ run during (**d**) years 11–150 and (**e**) years 151–224 with the forced signal removed (see Methods) and (**f**) their difference (i.e., years 151–224 minus years 11–150). Note the larger contour interval in (**f**) than that in (**c**). The stippling in (**c**) and (**f**) indicates that the SD difference is statistically significant at the 5% level based on a F-test (see Methods). A nine-point spatial smoothing was also applied in all panels.

thus NADW formation (Fig. 5c and Supplementary Fig. 11g–l). In other words, compared to the historical climate, weaker LHF anomalies (due to lack of the amplification from sea ice–air interactions) in the future climates are not able to generate large multidecadal anomalies in SSS and NADW formation, leading to much weaker and shallower multidecadal variability for AMOC, as shown by the 1%CO$_2$ run and CMIP5/CMIP6 simulations (Supplementary Fig. 18). Thus, as subpolar sea ice melts away, the lack of sea ice–air interactions would substantially weaken the multidecadal variability in the SIC-associated surface fluxes and upper-ocean conditions, leading to weaker and shallower AMOC variations in future warmer climates in CESM1 and other CMIP5/CMIP6 models.

Note that the mean climate and other conditions may change in the 23rd century in the CMIP5/CMIP6 model simulations or the CESM1 1%CO$_2$ run and it is difficult to isolate the impact of

the sea ice–air coupling in such fully coupled simulations, but the basic surface flux variations and their association with the SIC and AMOC fluctuations in these warming simulations are consistent with those seen in our CTL and CTL_FixedIce runs (Fig. 5c). Thus, the projected weakening of the AMOC's variability is consistent with the amplification effect of the sea ice–air interactions on multidecadal climate variations discussed above, although there are other changes in such warming simulations that prevent us from making definite conclusions regarding the role of sea ice–air coupling based on these simulations alone.

## Discussion
Our results, summarized in Fig. 8, suggest that although the poleward heat transport in the upper ocean from the northern

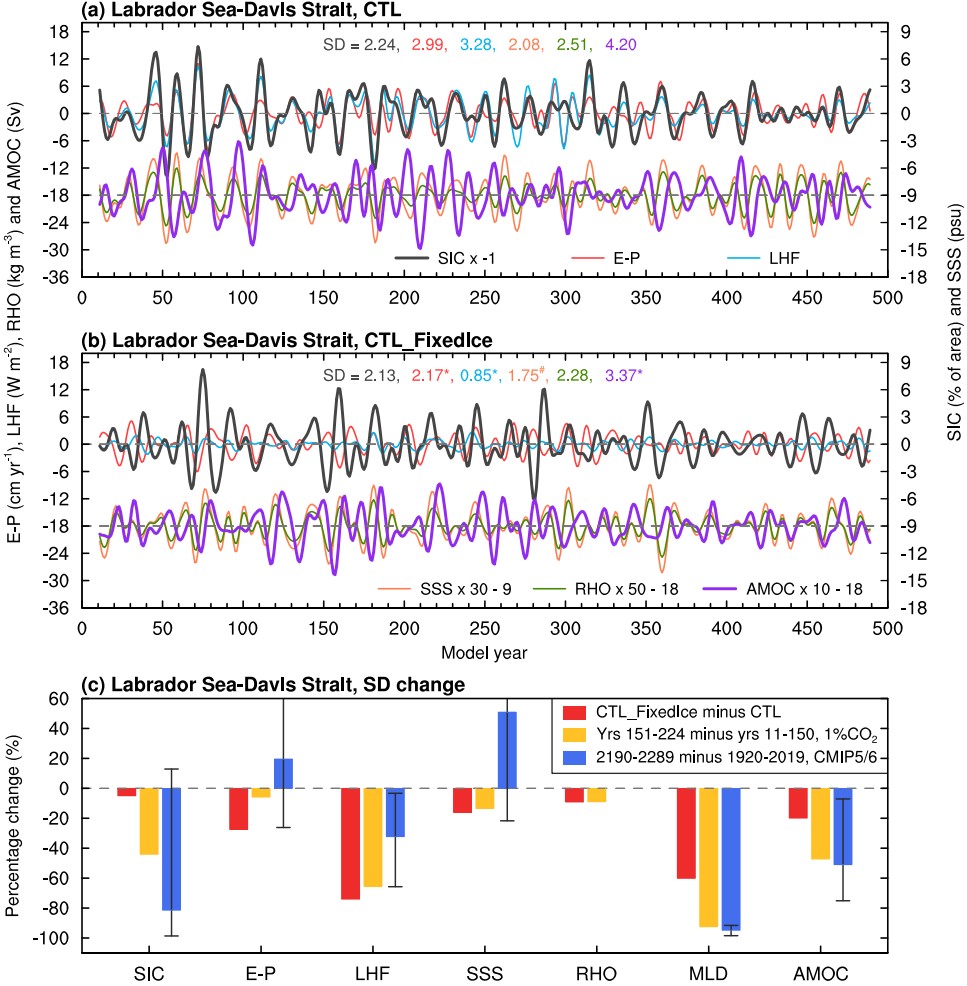

**Fig. 5 CESM1-simulated multidecadal fluctuations in regional sea-ice cover, AMOC and other associated oceanic variables.** Filtered time series of the DJF-mean anomalies of SIC (black, in % of area, sign reversed; right *y*-axis), surface evaporation-minus-precipitation (E–P; red, in cm year⁻¹; left *y*-axis), surface latent heat flux (LHF, positive upward; blue; in W m⁻²; left *y*-axis), upper-10m ocean salinity (SSS; orange, in psu, 1 psu = 1 g kg⁻¹; multiplied by 30 and shifted downward by nine to use the same right *y*-axis), and upper-10m ocean density (RHO; green; in kg m⁻³; multiplied by 50 and shifted downward by 18 to use the same left *y*-axis) averaged over the LSDS region (defined in Fig. 1b), and the AMOC index (purple; in Sv, multiplied by 10 and shifted downward by 18 to use the same left *y*-axis) from the CESM1 (**a**) CTL and (**b**) CTL_FixedIce run during years 11–490. A 10–90-year Lanczos band-pass filter was used. The SD of each curve is given in the respective color on each panel. The superscript "*" and "#" indicate the correlation (the SD difference) is statistically significant at the 5% and 10% levels, respectively, based on a resampling technique (a *F*-test) (see Methods). **c** Percentage changes of the SD of the multidecadal anomalies in SIC, E–P, LHF, SSS, RHO and MLD averaged over the LSDS region and the AMOC index from the CESM1 CTL_FixedIce minus CTL difference averaged over years 11–490 (red bars, relative to CTL), years 151–224 minus years 11–150 from the CESM1 1%CO₂ run (yellow bars, relative to years 11–150), and 2190–2289 minus 1920–2019 averaged over seven CMIP5 and CMIP6 models (blue bars, relative to 1920–2019; the whiskers denote the inter-model spread). Note that RHO data are unavailable and MLD data are only available for five CMIP5/6 models (see Supplementary Table 1).

North Atlantic to the GNS and BKS by the AMOC may trigger small multidecadal anomalies in SST and SIC over the subpolar Atlantic and Arctic regions[4,7,8], it is the local sea ice–air two-way interaction and the associated surface fluxes in the subpolar Atlantic and Arctic regions (including the LSDS, GNS and BKS), not the AMOC-induced heat transport itself, that are largely responsible for the large Tas and SIC multidecadal variations in these regions. Without the sea ice–air interactions, not only the Tas variations would largely disappear, but also the multidecadal variations in SIC and SSTs from the northern North Atlantic to BKS (and thus the poleward oceanic heat transport itself), as well as the AMV and AMOC, would weaken substantially. This is consistent with the important role of sea ice–air interactions for Arctic multidecadal climate variations suggested (but not clearly demonstrated) previously[31]. Because the AMV-induced SST variations are relatively small compared with the SST and Tas

variations over the LSDS, GNS and BKS in observations and the fully coupled CTL run, and a weakened AMOC and AMV cannot really affect winter surface fluxes over these regions due to fixed SIC in the CTL_FixedIce run, the weakened variability in CTL_FixedIce has to come mainly from the lack of the local sea ice–air coupling rather than the weakened AMOC.

The reduced AMOC multidecadal variability under weakened sea ice–air interactions is consistent with the projected future weakening of the AMOC multidecadal variability shown here and previously[29,30]. This suggests that winter sea ice–air interactions in the subpolar North Atlantic are a major mechanism for generating or amplifying multidecadal variability in AMOC and AMV, which is consistent with the significant roles of sea ice for AMOC and AMV suggested previously[13,14]. This differs from and complements previous notion that the AMOC's multidecadal variability is generated by stochastic atmospheric variability[32],

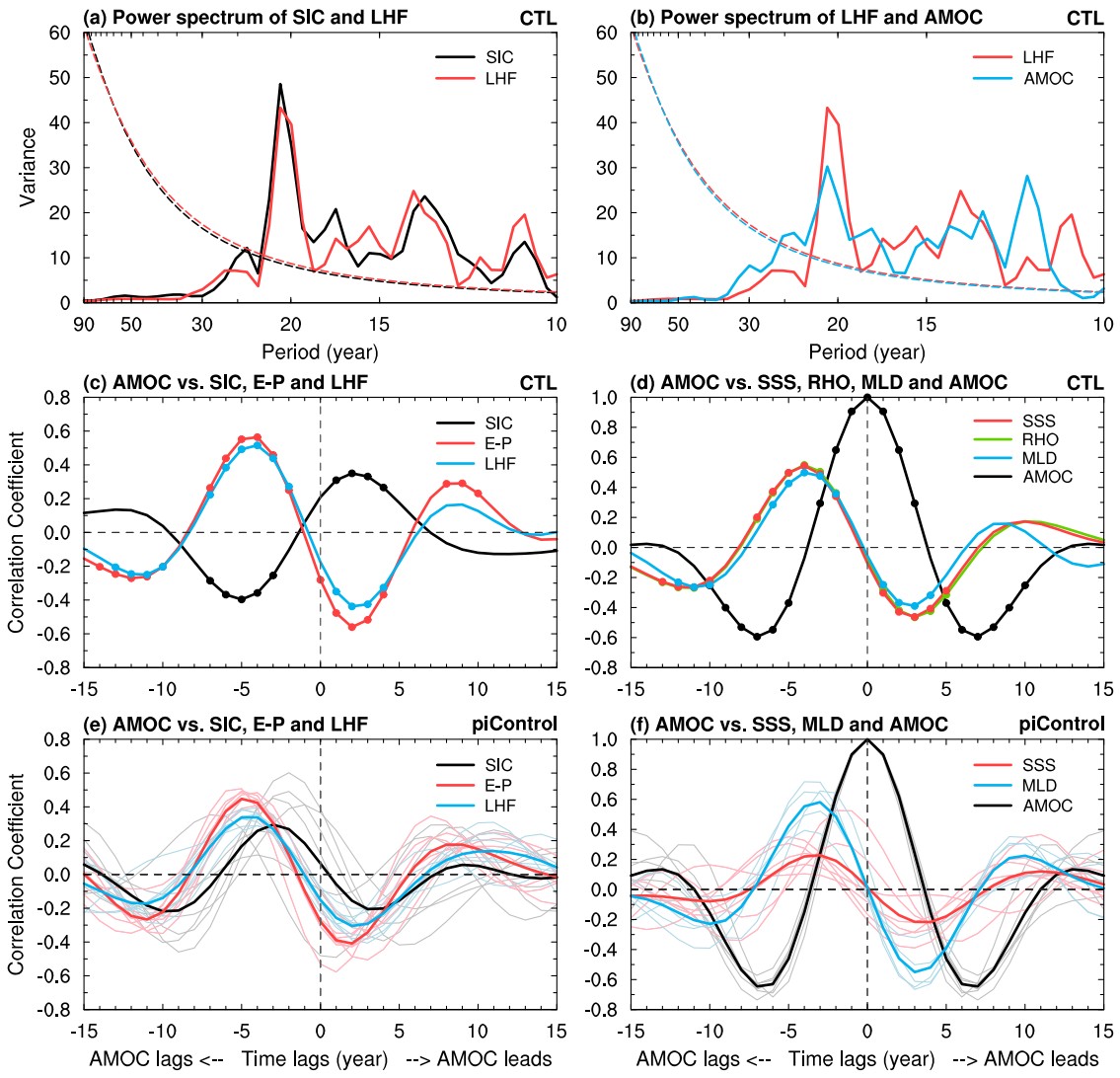

**Fig. 6 Relationship of regional sea-ice cover (SIC) with AMOC and other associated oceanic variables in CESM1. a, b** Power spectrum (standardized to use the same y-axis) of the DJF-mean 10–90-year band-pass filtered anomalies in (**a**) SIC and latent heat flux (LHF) averaged over the LSDS region (defined in Fig. 1b) and (**b**) LSDS LHF and AMOC index from the CESM1 CTL run as shown in Fig. 5a. The dashed lines are for the 95% confidence bound. **c, d** Lead-lag correlation coefficients of the DJF-mean band (10–90 year) filtered AMOC index with (**c**) SIC (black), evaporation minus precipitation (E–P, red), and LHF (blue) averaged over LSDS and with (**d**) sea surface salinity (SSS, red), surface ocean density (RHO, green), ocean mixed layer depth (MLD, blue) averaged over the LSDS, and AMOC itself (black) from the CESM1 CTL run as shown in Fig. 5a. The dots indicate the correlation coefficient is statistically significant at the 5% level based on the resampling technique (see Methods). **e, f** Same as (**c**), (**d**), but averaged over seven 500-year piControl simulations from seven CMIP5 and CMIP6 models (thin curves with the respective colors for individual model runs, and thick lines are the ensemble mean). Note that RHO data are unavailable and MLD data are only available for five CMIP5/6 models (see Supplementary Table 1).

oceanic planetary wave instability[33], thermohaline instability[34], and other processes[30,35]. Our new findings highlight the need to examine more closely the major impacts of the diminishing sea ice–air interactions in the LSDS, GNS, BKS, and other Arctic regions on Arctic and North Atlantic climate variability and on the AMV and AMOC under GHG-induced global warming, besides the weakening impact on AMOC's mean strength by Arctic sea-ice loss under increasing GHGs that results from upper ocean warming and freshening in the subpolar North Atlantic[10–12,16,26,27]. The long-term weakening of the AMOC under increasing GHGs, which concurs with declining sea ice, is due to the warming and freshening-induced upper ocean stratification, not due to the sea ice–air interactions discussed here, although the sea-ice loss can enhance the upper ocean stratification (e.g., through Arctic amplification of surface warming[22]) and thus contribute to AMOC's weakening[10–12].

In contrast to AMOC's weakening under GHG-induced warming with declining sea ice, for the internally generated decadal-multidecadal variations, reduced sea ice around the Labrador Sea and Davis Strait (e.g., due to a warm AMV phase) increases LHF and thus SSS and ocean density, which enhances the deep water formation and AMOC and thus amplifies the AMV. Apparently, this process plays a smaller role for AMOC's long-term response to GHG-induced warming than the warming and freshening-induced ocean stratification in climate change simulations with increasing GHGs, leading to an overall weakening of the AMOC under increasing GHGs and declining sea ice, in contrast to a stronger AMOC under lower SIC for internally generated multidecadal variations discussed above.

The AMOC's close link to Arctic sea ice suggests that models need to realistically simulate sea ice cover and its interactions with the atmosphere and oceans in order to reliably simulate the

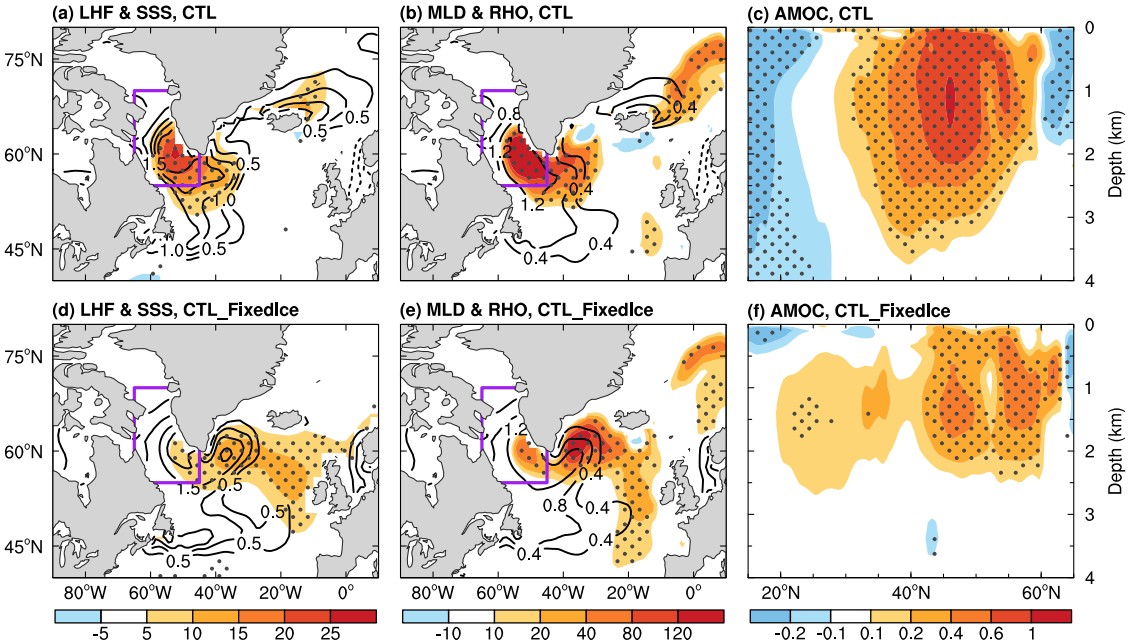

**Fig. 7 Composite differences associated with multidecadal latent heat flux (LHF) anomalies in CESM1. a–c** Composite differences of the 10–90-year band-pass filtered DJF-mean anomalies of (**a**) LHF (shading, positive upward, in W m$^{-2}$) and sea surface salinity (SSS, contours, in 0.1 psu), (**b**) ocean mixed layer depth (MLD, shading, in m) and surface ocean density (RHO, contours, in 0.1 kg m$^{-3}$), and (**c**) the 3–5-year lagged zonal-mean AMOC stream-function (shading, in Sv) between years with high (local maximum higher than +1SD) and low (local minimum lower than –1SD) LHF (i.e., high LHF years minus low LHF years) over the LSDS region outlined in (**a**), (**b**) as in Fig. 1b) based on the CESM1 CTL run from years 11 to 490. (**d–f**) Same as (**a–c**) but based on the CESM1 CTL_FixedIce run. A nine-point spatial smoothing was also applied in all panels. The stippling indicates that the difference is statistically significant at the 5% level based on a Student's *t*-test.

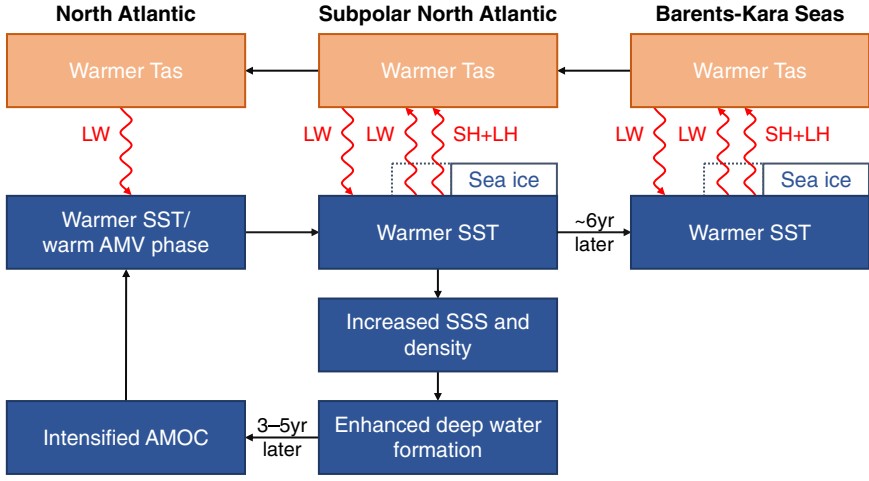

**Fig. 8 Schematic diagram for the amplification of the multidecadal variability by sea ice–air interactions.** Here two-way sea ice–air interactions are triggered by a multidecadal sea-ice loss (indicated by the dashed lines) under a warm Atlantic Multidecadal Variability (AMV) phase. AMOC is for Atlantic Meridional Overturning Circulation, LW is for longwave radiation, SH is for surface sensible heat flux, and LH is for surface latent heat flux. Red color indicates that the fluxes are increased. The key processes shown here include (1) a warm SST anomaly over the subpolar North Atlantic is advected to subpolar ice margins and causes large multidecadal Tas and SIC variations along the ice margins through surface upward energy fluxes (red upward vectors), which in turn warms the subpolar North Atlantic and Arctic surface mainly through downward LW radiation (red downward vectors); and (2) a multidecadal sea-ice loss over the subpolar North Atlantic, especially over the Labrador Sea, would increase LH flux, upper-ocean salinity and density and thus deep water formation, leading to a deeper and stronger AMOC in ~3-5 years (i.e., the AMOC lags the SST anomalies and thus AMV by 3–5 years), which further enhances poleward heat transport to enlarge the warm NASST anomaly. The processes are reversed during a positive multidecadal sea-ice anomaly induced by a cold AMV phase.

responses of the AMOC and other associated fields to future GHG increases. Our results suggest that biases in model-simulated AMOC strength for the current climate[6] could be related to biases in model-simulated mean sea-ice extent over the Labrador Sea and Nordic Seas, as excessive or too little ice cover

in these regions could lead to unrealistic sea ice–air interactions, leading to unrealistic SSTs and SSS and thus deep water formation. Further work is needed to examine how the current AMOC strength and its future change are linked to current sea ice biases and future sea-ice loss in the Atlantic sector. Correlations of the

decadal-multidecadal variations between AMOC and LSDS SIC reveal considerable differences among different models for both piControl runs (Fig. 6e, f and Supplementary Fig. 12) and historical simulations (Supplementary Fig. 17), suggesting substantial uncertainties in current model simulations of this crucial relationship, as noticed previously[36]. Further investigations of the SIC-AMOC relationship in other climate models are needed. We noticed substantial uncertainty in the connection of BKS SIC and Tas variations to the AMV simulated by the CESM1 (Supplementary Fig. 3d) and other climate models (not shown), but this deficiency should not affect our main conclusion on the critical role of sea ice–air coupling for local SIC and Tas variability, AMV and AMOC, which is not significantly affected to BKS sea ice–air coupling.

As subpolar sea ice is projected to decrease[15,16], the sea-ice margins and thus the sea ice–air interaction centers retreat poleward under GHG-induced warming, which is consistent with the reported northward shifts of the surface flux forcing of the AMOC[29] and the poleward movements of the deep water formation regions[37], although the zonal-mean ocean mixed layer depth did not move northward in CMIP5 models[30]. Clearly, the enhanced sea ice–air interactions at higher latitudes are not enough to compensate the lost forcing over the current ice margins and from the upper-ocean freshening and other processes in the subpolar North Atlantic, leading to weakened AMOC and its variability under global warming. More work is needed to show how the northward shifts of the sea ice–air interactions affect AMOC and AMV under GHG-induced warming. Furthermore, models show a slow recovery of the AMOC after $CO_2$ stabilizes[38] or after a rapid summer sea ice loss[39], but it is unclear whether the reduced variability of the AMOC and AMV would recover due to other processes in an equilibrium warmer climate when subpolar North Atlantic sea ice and thus sea ice–air interactions disappear.

## Methods

**Observational and CMIP data.** Monthly mean surface air temperature (Tas), sea surface temperature (SST) and sea ice data from 1950 to 2020 were obtained from the newly released European Centre for Medium-Range Weather Forecasts (ECMWF) Reanalysis version 5 (ERA5)[40] on a 1° grid. The Tas results are similar using the NCEP/NCAR reanalysis except for stronger variability over the three regions outlined in Fig. 1a. We focus on the boreal winter, defined as December–January–February (DJF; e.g., the winter of 1950 is for December 1950–February 1951; thus, our last winter is for 2019 from December 2019–February 2020); however, we also examined other seasons. The results for annual mean (cf. Supplementary Fig. 19) and other seasons are similar except with smaller magnitudes.

We also used monthly model data from twelve all-forcing historical (HIST) and RCP8.5 (RCP8.5 for CMIP5; SSP5-8.5 for CMIP6) simulations from twelve CMIP5[41] or CMIP6[42] models (Supplementary Table 1). These models had simulations extended to year 2300 under the extended RCP8.5 scenario, allowing us to investigate the multidecadal variability over the North Atlantic and Arctic in an approximately ice-free world. The purpose of using the 23rd century simulation is to find out what would happen when most of the Arctic ice is gone. Eight of the models were from the CMIP5 as follows: bcc-csm1-1, CCSM4, CNRM-CM5, GISS-E2-H, GISS-E2-R, HadGEM2-ES, IPSL-CM5A-LR, and MPI-ESM-LR. Four of the models were from the new CMIP6: CanESM5, CESM2-WACCM, IPSL-CM6A-LR, and MRI-ESM2-0. We also used 500-year pre-industrial control (piControl) simulations from seven CMIP5/CMIP6 models with the AMOC data available to us, namely, CCSM4, CNRM-CM5, GISS-E2-R, MPI-ESM-LR, CESM2-WACCM, IPSL-CM6A-LR, and MRI-ESM2-0. Before further analysis, all the monthly model outputs were remapped onto a 2.5° grid for atmospheric fields and a 1° grid for ocean or sea-ice fields.

**CESM1 simulations.** We used the Community Earth System Model version 1.2.1 (CESM1)[43] from the National Center for Atmospheric Research with version 4 of the Community Atmosphere Model (CAM4) for its atmospheric component. The CESM1 has been widely used to study the global and Arctic climate and it simulates the global, Arctic and midlatitude mean climate fairly realistically, including the spatial and seasonal patterns of the sea-ice and surface fluxes and their interannual variability[22,44,45]. We ran the CESM1 with grid spacing of 2.5° longitude × ~2.0° latitude for the atmospheric model, and ~1.0° longitude × ~0.5°

latitude for the sea-ice and ocean models. Most of the simulations used here (including the fixed-ice setup) have already been described in detail and used by refs. [21,22], and [45].

We made two types of long-term CESM1 simulations: one type with fully coupled dynamic sea ice (i.e., with the sea ice–air two-way interactions) and the other type (FixedIce) with fixed sea ice concentration (SIC) only in the coupler of the model for estimating the water and ice fractions used in calculating grid-box mean values of the exchange fluxes of energy, mass and momentum between the atmosphere and underlying ocean and ice surfaces. For each type, we made one 500-year pre-industrial control (piControl) run with atmospheric $CO_2$ fixed at 284.7 ppmv and one 235-year simulation with increasing $CO_2$ by 1% per year (1% $CO_2$). The FixedIce 1%$CO_2$ simulation (1%$CO_2$_FixedIce) is described in detail and used by refs. [21,22], and [45]. The FixedIce runs, which include a new 500-year piControl run with fixed SIC (referred to as CTL_FixedIce), use fixed SIC derived from the monthly climatology from the standard piControl run (CTL) only in the coupler for determining the ice and water surface fractions that are used as the weights in calculating the grid-box mean fluxes. The fluxes over the ice or water fraction, as well as the sea-ice cover within the sea-ice model, are not altered by us. Only their weights may be affected by our prescribed SIC (when it differs from the internal SIC, which occurs mainly around the ice margins). Thus, the use of a fixed SIC in the coupler, albeit unnatural, does not alter any physical laws (including the conservation of energy, mass and momentum).

By fixing the SIC to its climatological value in our fixed-ice runs, instead of using its internal value as in the standard CESM1, we essentially cut off the impact of a variable SIC on the climate. This is similar to using climatological SST instead of time-varying SST from observations in AMIP-type simulations, except in our case we have a fully coupled ocean and ice model to feel the impact from the fixed SIC used in the coupler. One can also think of our fixed-ice setup as a change of ocean surface type (between water and ice) for certain grid boxes and at certain times (when the internal SIC differs from its climatological SIC), similar to changing a land surface type (e.g., from forest to croplands) for studying the effect of the land cover change (although this change is permanent and over solid land over the specified grid boxes while our change occurs over liquid ocean only when the internal SIC differs from its climatology—if the two SIC values are the same, e.g., near the North Pole where both SIC are 100% at most of the time, then there is no change at all from our setup). While both changes are unnatural and artificial, such modeler-specified surface type changes (e.g., urban surfaces, croplands) are already included in the standard CESM1; thus, we view our prescribed SIC as another modeler-induced perturbation of the climate system for studying its impact on the climate.

In our FixedIce runs, we did not attempt to fix sea ice in the ice model, in contrast to previous similar simulations[12,44,46–48]; rather, we specifically cut off the impact of sea ice variations on the exchange fluxes between the atmosphere and the ocean/ice surface while allowing the atmosphere and ocean to influence sea ice freely. Thus, we can attribute any differences between the CTL and CTL_FixedIce runs or between the standard 1%$CO_2$ and the FixedIce 1%$CO_2$ runs to a specific physical process, namely the impact of varying sea ice through changes in surface fluxes. That is, all the internal processes (including the impact of AMOC's advection on North Atlantic surface conditions) are allowed to work in both simulations, except for the sea ice–air two-way interactions that are missing in the FixedIce run. In other words, these differences are triggered by the lack of the two-way ice–air interactions in the FixedIce run. We will refer the sea ice–air (and sea ice-ocean) interactions in the standard CTL and 1%$CO_2$ runs as two-way interactions, while they are referred to as one-way interaction (i.e., air-to-ice and ocean-to-ice only) in the CTL_FixedIce and 1% $CO_2$_FixedIce runs.

Here, we focus on the decadal-multidecadal variations, rather than long-term mean changes in response to $CO_2$ forcing[12,44,46–48] over the Arctic and North Atlantic regions in individual realizations, which resemble the real world.

**Decadal to multidecadal variability.** As our focus is on internally generated decadal-multidecadal variability, we first removed the externally forced signal in all analyzed fields from reanalysis and model simulations based on the method used previously[49,50]. Here we defined the forced signal as the time series of the global-mean (60°S–75°N) surface air temperature (GMT) averaged over twelve all-forcing historical and RCP8.5 runs from twelve CMIP5/CMIP6 models used in this study. For reanalysis over 1950–2020 and CMIP5/CMIP6 historical and RCP8.5 runs over 1900–2299, we used linear regression between the CMIP model-averaged GMT (as the x variable) and a given field (as the y variable) at each grid box to obtain the forced component in variable y, and then subtracted this forced component from the original time series of variable y. The regression coefficient varies spatially and largely accounts for different regional responses to the same forcing[49]. Similar procedure was also applied to the CESM1 1%$CO_2$ and 1%$CO_2$_FixedIce runs from years 1–235 but using their own GMT time series as the x variable in a 3rd-order polynomial fit in the detrending procedure (instead of the linear fit) to eliminate any nonlinear long-term trend, as Arctic sea-ice concentrations decrease nonlinearly with monotonically increasing $CO_2$. After removing the externally forced signal (except for the CESM1 and CMIP5/CMIP6 pre-industrial control simulations), the internally generated decadal-multidecadal anomalies were obtained by applying a 10–90-year Lanczos band-pass filter with 21 weights, to filter out any

interannual or centennial-multicentennial variability in order to focus on the decadal-multidecadal variability. Note that results are similar using more filter weights (e.g., 181 weights) but with larger multidecadal magnitudes and, more importantly, a greater data loss (and thus a smaller sample size, especially for observations and CMIP5/6 all-forcing simulations).

We calculated the standard deviation (SD) of the detrended and filtered DJF-mean anomalies to quantify the decadal-multidecadal variability. Note that the SD may not be sufficient for quantifying the probability density function, which is not our purpose here, as the distribution may not be exactly Gaussian. For the CMIP5/CMIP6 simulations, the SD pattern of the detrended and filtered DJF-mean anomalies was first obtained from each simulation and then averaged over the models (one simulation for each model) with equal weighting to create the multi-model mean pattern.

**AMV and AMOC indexes**. The AMV index was defined as the detrended and filtered DJF-mean SST anomalies averaged over the northern North Atlantic (0°–60°W, 50°–65°N), and the AMOC index was defined as the detrended and filtered DJF-mean anomalies of the Atlantic zonal-mean meridional stream function averaged over the depths of 1000–2000 m and 40°–55°N throughout the study, as these two regions see larger multidecadal variability in SST (Figs. 1 and 2) and AMOC stream function (Fig. 4), respectively.

**Statistical significance test**. We used $F$-tests to test the significance of a change in a given variable's standard deviation, with the effective degree of freedom defined as $N/\tau$ (where $N$ is data length and $\tau$ is the $e$-folding time scale over which the autocorrelation decays to $1/e$). Student's $t$-tests were applied to test whether the composite differences are statistically significant based on a 5% significance level. The significance of a correlation between two strongly autocorrelated time series ($X$ and $Y$ of length $N$) was tested on the basis of a resampling method[51]: we randomly selected a number $i$ between 1 and $N$ to reconstruct a new $X$ that starts from year $i$ to year $N$ and followed by the original $X$ from year 1 to year $i−1$ immediately, and a similar reshuffling process was applied to $Y$ but with a different random number $i$. We calculated the correlation between the randomly reshuffled $X$ and $Y$, and then repeated this resampling procedure for 10,000 times to create a correlation distribution that could occur by chance. For a 5% significance level based on a one-tailed test, the 5th and 95th percentile values of this correlation distribution were chosen as the lower and higher confidence bounds, respectively.

## Data availability

All observational data used in this study are publicly available and can be downloaded from the corresponding websites. ERA5: https://www.ecmwf.int/en/forecasts/datasets/reanalysis-datasets/era5; NCEP/NCAR reanalysis: https://psl.noaa.gov/data/gridded/data.ncep.reanalysis.html; The CMIP model data used in this study can be obtained from the CMIP5 and CMIP6 archives at https://esgf-node.llnl.gov/projects/esgf-llnl/. The CESM1 model data used in this study are available from the authors upon request.

## Code availability

The code of the CESM1 model used in this study is available from http://www.cesm.ucar.edu/models/cesm1.2/.

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

## Acknowledgements

J.D. was supported by the National Natural Science Foundation of China (grant nos. 42088101 and 41705054). A.D. was supported by the National Science Foundation (grant nos. AGS-2015780 and OISE-1743738).

## Author contributions

A.D. formulated the main ideas, contributed to the design of the analyses and figures, wrote the first draft of the paper, and designed the CESM1 simulations and made all the CESM1 simulations except CTL_FixedIce, which was ran by J.D. J.D. contributed to the main ideas, did all the analyses, made all the figures, and helped writing the manuscript.

## Competing interests

The authors declare no competing interests.
