## [Peer Review File · Nature Communications]

Sea ice-air interactions amplify multidecadal variability in the North Atlantic and Arctic regionReviewers' Comments:

Reviewer #1:

Remarks to the Author:

Review of Sea ice-air interactions amplify multidecadal variability in the North Atlantic and Arctic region by Deng and Dai

The paper discussed the mechanism by which sea ice-air interactions can amplify the multidecadal variability in the North Atlantic and Arctic region. It is shown that multidecadal variations in those regions are greatly reduced when sea ice is fixed in the idealized simulations that decouple sea ice-air interactions or sea ice melts away in CMIP model projections under GHG-induced warming. Thus, the authors concluded that sea ice-air interactions are crucial for the multidecadal climate variability in the North Atlantic and Arctic regions. I found that the results presented in the paper very interesting, which could contribute to the debate on the role of sea ice in the recent decadal trend in Arctic temperature and related midlatitude impacts (if possible). However, I also feel that despite many figures being presented in the paper, the paper could benefit strongly from more direct assessments of the role of sea ice in the observed temperature trend in recent decades.

Major comments:

1. The paper was motivated by rapid Arctic warming since the late 1990s, and the paper discussed the role of sea ice in the multidecadal variability with a peak of about 20 years in the power spectrum (Fig. 3). However, I was surprised that the paper did not discuss the implications of the results for the Arctic warming trend from 1990 to present, which has been a focus in the literature. There are some recent studies on the weakened connection between the Arctic and midlatitudes (Blackport & Screen, 2020), which may be related to the multidecadal cycle. More importantly, while the authors cite Blackport et al., (2019) for the importance of internal variability, their paper emphasizes the role of the atmosphere in driving the sea ice variability (negative heat flux near the ice melting regions), whereas this study highlights the role of sea ice (positive heat flux in the sea ice melting regions) in driving the multidecadal variability. I understand the two studies are focused on different time scales (interannual vs decadal), but I think the clarification on the role of observed sea ice in temperature variability in recent decades would be critical to the ongoing debate on sea ice and decadal variations in Arctic temperature.

2. The paper compares the reduction in multidecadal variability for the two scenarios, one in which sea ice is fixed in the idealized simulations that decouple sea ice-air interactions, and the other in which sea ice melts away in CMIP model projections under GHG-induced warming. However, the comparison is qualitatively at best in my opinion since many processes are involved in the climate warming projections such as the change in stratification. Because the mean AMOC is projected to weaken in a warming climate, it is not surprising that the variability will decrease in amplitude along with the weakening in mean amplitude. I think the authors should use more quantitative metrics to estimate the magnitude of weakening with respect to sea ice and other factors and compare the changes for different scenarios explored in this study. Some of the changes in variability are scattered in different figures, but this could be synthesized in a quantitative way (e.g., a bar chart for different factors in the present climate and a sea-ice free climate). I think a quantitative summary for the role of sea ice based on different experiments is needed to better support the conclusion of this paper.

3. While the text is reasonably concise, I feel there are too many figures for the audience of a typical nature style article. The present manuscript includes 10 Figures in the main text, 10 Figures in the Extended Data, and 10 Figures in the supplementary information. I would recommend the paper to a more technical journal in climate sciences if there is too much information in the paper that cannot be synthesized for a general audience.

Other minor comments:

L24-25: can you include estimates of the reduction in variability for all the variables here?

Fig. 2: the caption says that only negative values are plotted in c and f, but it is likely misleading to only plot the regions of decrease in SD, not the regions of increase in SD. I think it should be left to readers to see/decide whether the region of increase is unimportant or not. The same comment applies to the stream functions in Fig. 4c and Fig. 8c

Method to fix SIC: in my understanding, the paper uses a method to override the water and ice fractions within a grid-box in the coupler module of a coupled ocean-atmosphere model. While this is certainly a new method to decouple the sea ice-air interaction, I am not sure about the claim that the method does not alter any physical laws (L404-405) --- shouldn't this alter the amount of ice or water used in the flux calculation? The authors further explain that this is in analogy to the change of land surface type (L410-415). While this is explained in the method section, I think brief explanations in the main text would help readers understand the physical meaning of fixing sea ice in flux calculation.

References

Blackport, R., & Screen, J. A. (2020). Weakened evidence for mid-latitude impacts of Arctic warming. *Nature Climate Change*, 10(12), 1065–1066. <https://doi.org/10.1038/s41558-020-00954-y>
Blackport, R., Screen, J. A., van der Wiel, K., & Bintanja, R. (2019). Minimal influence of reduced Arctic sea ice on coincident cold winters in mid-latitudes. *Nature Climate Change*, 9(9), 697–704. <https://doi.org/10.1038/s41558-019-0551-4>

Reviewer #2:

Remarks to the Author:

I find this study very interesting and well-written and the authors' methodology of fixing sea ice via fluxes through the coupler is novel and has previously contributed to the understanding of Arctic Amplification (Dai et al, 2019). Overall, I believe this paper is suitable for publication; however, I am hoping that the authors can clarify a few issues prior to publication. I have three major comments and a few minor comments:

Major Comment 1:

I find the results very interesting, but at the same time I am struggling a bit with the chicken-and-egg nature of the results. Let's start with multi-decadal sea-ice variability in the LSDS region. The authors argue that this can affect AMOC variability and show that this is the case in their FixedIce runs and in the warming runs after sea-ice has declined in this region. The authors also argue that multi-decadal variability in the BKS region is diminished because of the lack of coupling. But, if variability in the AMOC is diminished, could this not be affecting the variability in the BKS region alone, rather than it being because of the lack of coupling? Is it possible for the authors to fix the sea-ice fluxes just in the LSDS region to see to what extent the variability in the BKS region is affected?

Major Comment 2:

Here is an alternative view. The authors argue that the AMV affects BKS sea-ice and the two-way coupling is required to translate into multi-decadal variability in BKS SIC and Tas. However, Figure ED 2d suggests that this connection does not really exist in CESM1 - the correlations are quite weak compared to the reanalysis. So, is it possible that multi-decadal variability in BKS SIC and Tas in CESM1 is simply a local coupled ice-atmosphere mode and not connected to the AMV?

Major Comment 3:

There are two instances in particular where CESM1 seems to behave either differently from reanalysis (Fig. ED 2d) or differently from other GCMs/ESMs (Fig. 6c versus d). These differences affect the

interpretation of the results. I realize the CESM1 is a well-documented and reasonably well-performing model, but I am hoping the authors can provide some clarification or reasoning behind why these differences are acceptable. For example, can the authors reproduce Fig. ED 2d for the CMIP5/CMIP6 models as a point of comparison?

Minor Comments:

Abstract: The wording of the first sentence of the abstract is a bit confusing. It reads as though the authors are saying that the rapid warming of the Arctic is due to internal variability. Of course, much of that signal is a forced anthropogenic signal. So, I believe the authors are just referring to a multi-decadal component. I suggest rephrasing to make this clearer.

Line 32: There is some literature that relates multi-decadal variability in the Arctic to forcings other than GHGs, i.e. aerosols (Shindell and Faluvegi, 2009; Fyfe et al. 2013). This forced multi-decadal signal is much larger in the Arctic than the global mean. When the authors detrend based on global mean SAT, to what extent do the aerosol effects on Arctic warming remain as part of the multi-decadal signal (this is only an issue for the reanalysis and historical model runs)?

Lines 141-153: This paragraph is based on the argument that BKS sea ice is affected by AMV SST anomalies, but Fig. ED 2d does not really show this for CESM1.

Lines 225-234: The lag relationships are quite different in CESM1 compared to CMIP5/CMIP6 - can the authors explain? Why should the reader accept that CESM1 is realistic?

Lines 242-246: wouldn't the enhanced LHF also lead to a local cooling and sea ice growth? In Fig. 6c, when AMOC leads, there is a positive correlation between SIC and AMOC. I am not exactly sure how this fits into the overall interpretation of the results presented in the schematic in Figure 10?

RESPONSE TO REVIEWER COMMENTS ON

Sea ice-air interactions amplify multidecadal variability in the North Atlantic and Arctic region by Deng and Dai

Reviewer #1 (Remarks to the Author):

The paper discussed the mechanism by which sea ice-air interactions can amplify the multidecadal variability in the North Atlantic and Arctic region. It is shown that multidecadal variations in those regions are greatly reduced when sea ice is fixed in the idealized simulations that decouple sea ice-air interactions or sea ice melts away in CMIP model projections under GHG-induced warming. Thus, the authors concluded that sea ice-air interactions are crucial for the multidecadal climate variability in the North Atlantic and Arctic regions. I found that the results presented in the paper very interesting, which could contribute to the debate on the role of sea ice in the recent decadal trend in Arctic temperature and related midlatitude impacts (if possible). However, I also feel that despite many figures being presented in the paper, the paper could benefit strongly from more direct assessments of the role of sea ice in the observed temperature trend in recent decades.

Response: We thank the reviewer for the generally positive review and the constructive comments, which are in black. Our response is in blue. New text added to the manuscript is in red.

Major comments:

1. The paper was motivated by rapid Arctic warming since the late 1990s, and the paper discussed the role of sea ice in the multidecadal variability with a peak of about 20 years in the power spectrum (Fig. 3). However, I was surprised that the paper did not discuss the implications of the results for the Arctic warming trend from 1990 to present, which has been a focus in the literature. There are some recent studies on the weakened connection between the Arctic and midlatitudes (Blackport & Screen, 2020), which may be related to the multidecadal cycle. More importantly, while the authors cite Blackport et al., (2019) for the importance of internal variability, their paper emphasizes the role of the atmosphere in driving the sea ice variability (negative heat flux near the ice melting regions), whereas this study highlights the role of sea ice (positive heat flux in the sea ice melting regions) in driving the multidecadal variability. I understand the two studies are focused on different time scales (interannual vs decadal), but I think the clarification on the role of observed sea ice in temperature variability in recent decades would be critical to the ongoing debate on sea ice and decadal variations in Arctic temperature.

Response: We thank the reviewer for this thoughtful comment. We agree that the multidecadal variability induced by sea ice-air interactions are highly relevant for the rapid warming in the Arctic since the 1990s, and we plan to examine this issue thoroughly in a separate study.

We showed in a separate study (Dai and Deng 2021) that Arctic sea ice-air interactions have contributed to the enhanced warming over the Barents-Kara Seas (BKS) and the winter cooling over central Eurasia from 1992–2012 and the two regional temperature decadal trends are connected via cold advection by increased Ural blocking associated with the enhanced BKS warming. We also showed that the variability of decadal temperature trends over the BKS region is amplified by the sea ice-air interactions (see red curves in Fig. R1, which is taken from Dai and Deng 2022), and Fig. R2 shows that the recent BKS warming trend from 1992–2012 is more likely to occur with the sea ice-air coupling than the case without the coupling. These results imply that the recent decadal BKS warming trends may partly result from the sea ice-air interactions. However, we have not thoroughly examined the contribution of the multidecadal variability induced by the sea ice-air interactions to decadal trends in Arctic mean temperature or over other Arctic regions. We plan to investigate this issue in a separate study by comparing the decadal trend differences in our CTL and CTL_FixedIce runs over different regions of the high latitudes, and discuss the implication for the observed warming trends in the Arctic during recent decades. In the present study, our focus is on the role of Arctic sea ice-air interactions in generating multidecadal variability in the Arctic and North Atlantic, especially for AMOC and AMO, which play a major role for regional and global climates.

Blackport et al. (2019) cleverly used the direction of the surface total heat flux (THF) anomaly and the lead-lag correlation of monthly variations to infer whether the atmosphere or sea ice is the driver of the concurring variations. Their Fig. 3 seems to suggest that the correlation is strongest or similar with zero lag in ERA-Interim and EC-Earth, implying the simultaneous nature of the interactions on monthly and longer time scales. We also wonder if their CBS THF values (their Fig. 1) are representative for other Arctic regions (e.g., the BKS) and how often their case (d) occurs during a typical winter. If most of the winter days are dominated by their case (c) or (a), then the DJF mean will be dominated by oceanic influence, which is the cause for GHG-induced change (Dai et al. 2019). If case (d), i.e., negative THF over ice melting region, occurs only over a small number of the days during a typical winter or dominates only in a small number of the total winters over most Arctic ice margin zones, then one cannot conclude that the atmosphere drives interannual sea ice variations in the Arctic. It appears that Blackport et al. (2019) only examined the CBS region, not the BKS or LSDS regions examined in our study. Because of these issues, we are unable to compare our findings with Blackport et al. (2019) regarding which (sea ice or the atmosphere) is the driver of the concurring variations.

In response to this comment, we have added the following text to the end of the Summary and Discussion section (around lines 364–370):

“The multidecadal temperature variations induced by the sea ice-air interactions can cause apparent warming or cooling trends over multidecadal periods over many Arctic regions, such as the BKS, where rapid warming was observed from 1992–2012 that is attributed partly to the

sea ice-induced multidecadal variability and also contributes to the concurring winter cooling over Eurasia via atmospheric cold advection⁶. However, how much of the rapid warming in the Arctic during recent decades can be attributed to the multidecadal variability induced by the sea ice-air interactions needs further investigation.”

6. Dai, A., & Deng, J. Recent Eurasian winter cooling partly caused by internal multidecadal variability amplified by Arctic sea ice-air interactions. *Climate Dynamics*, <https://doi.org/10.1007/s00382-021-06095-y> (2022).

Fig. R1. Time series of 21-year running trends of the 9-year low-pass filtered DJF-mean Tas anomalies averaged over the BKS from the CESM1 (a) CTL run and (b) CTL_FixedIce run during years 1–500. The standard deviation (SD) of each curve is given on the panel and the “*” indicates the SD difference of the two curves is statistically significant at the 5% level based on a *F*-test. (Same as the red curves shown in Fig. 4 of Dai and Deng 2022)

Fig. R2. The probability density function (PDF) of BKS surface air temperature (Tas) trends over all 21-year moving periods from the CESM1 standard (CTL) and FixedIce (CTL_FixedIce) control simulations (as shown in Fig. R1). The x axis is the 21-year trend and the y axis is the occurrence frequency (in %). A fixed bin number of 52 was used, and a five-point averaging was applied on all curves for clarity. The standard deviations (SD) of the distributions are also given in parentheses with the same colors. The stars indicate, from left to right, the BKS cooling (from 1971–1991) and warming (from 1992–2012) trend from ERA5. The vertical solid (dashed) lines are, from left to right, the 5th and 95th percentile values in the CESM1 CTL (CTL_FixedIce) run; and the dots on their top ends indicate the percentile values are significantly different between these two runs at the 5% level on the basis of a resampling technique, which as used to test whether the 5th or 95th percentile values of two time series (X of length N_1 and Y of length N_2) are significantly different from each other: the concatenated time series (length of $N_1 + N_2$) of the original X and Y is randomly sampled to generate a new pair of X (length of N_1) and Y (length of N_2); we calculated the difference of the 5th or 95th percentile values between the resampled X and Y and then repeated this procedure for 10,000 times to create a distribution of these differences that could occur by chance. For a 5% significance level based on a two-tailed test, the 2.5th and 97.5th percentile values of such a difference distribution were chosen as the lower and higher confidence bounds, respectively. (Similar to Fig. S7 of Dai and Deng 2022).

2. The paper compares the reduction in multidecadal variability for the two scenarios, one in which sea ice is fixed in the idealized simulations that decouple sea ice-air interactions, and the other in which sea ice melts away in CMIP model projections under GHG-induced warming. However, the comparison is qualitatively at best in my opinion since many processes are involved in the climate warming projections such as the change in stratification. Because the mean AMOC is projected to weaken in a warming climate, it is not surprising that the variability will decrease in amplitude along with the weakening in mean amplitude. I think the

authors should use more quantitative metrics to estimate the magnitude of weakening with respect to sea ice and other factors and compare the changes for different scenarios explored in this study. Some of the changes in variability are scattered in different figures, but this could be synthesized in a quantitative way (e.g., a bar chart for different factors in the present climate and a sea-ice free climate). I think a quantitative summary for the role of sea ice based on different experiments is needed to better support the conclusion of this paper.

Response: As the reviewer correctly pointed out (and also stated in the revised manuscript around lines 184–187 and 278–280), the 23rd century climate contains many other changes that complicate the interpretation of the AMOC changes. Without a parallel idealized simulation with fixed sea cover (like our FixedIce run), it is difficult, if not impossible, to isolate and quantify the role of sea ice-air interactions from the effect of other changes (e.g., the weakening of the AMOC) in such fully coupled simulations with large GHG increases. Thus, the 23rd century results were used only as a consistency check with our CESM1 results. Please note that we also examined the multidecadal relationships between SIC and AMOC in piControl runs by other climate models (new ED Fig. 6 and Fig. 6e).

For the 23rd century simulations, besides the AMOC change, we also showed the latent heat flux (LHF) change (new Fig. S4) and compared the LHF change with SST's variability change (new ED Fig. 4b). These results suggest that without the SIC-induced large LHF variations through the sea ice-air coupling, the multidecadal NASST variations would become much weaker in the 23rd century. As pointed out around lines 260–264, while we expect AMOC's variability to weaken as its mean strength decreases, the surface flux changes associated with reduced sea ice-air interactions (not with the weakening of the AMOC) would also weaken AMOC's variability in the 23rd century. Thus, the weakening mechanism for AMOC's variability in the 23rd century in CMIP models is qualitatively consistent with the underlying process we found in the CESM1, as stated around lines 278–282.

Furthermore, while the mean strength of the AMOC is expected to decrease, which could lead to reduced AMOC variations (as pointed out around lines 260–262), under GHG-induced global warming due to increased ocean stratification in the subpolar North Atlantic, what causes the AMOC to weaken under increasing GHGs is still investigated. Our recent work (Dai 2022) shows that the enhanced Arctic warming caused by sea-ice loss is largely responsible for the continued weakening of the AMOC under large CO₂ increases (now pointed out around lines 257–258) mainly due to increased runoff from Greenland and decreased oceanic evaporation caused by reduced ocean-air temperature gradients in simulations with large Arctic amplification and sea-ice loss. In other words, the weakening of the AMOC is also linked to sea-ice loss (and thus the lack of sea ice-air interactions) under large CO₂ increases.

As explained above, unlike our pair of CESM1 simulations with and without the sea ice-air coupling, the CMIP fully coupled simulations do not allow us to cleanly isolate and quantify the impact of the sea ice-air coupling on the AMOC and other fields (which is now pointed out around lines 279–280). Thus, we focused on the underlying processes in these CMIP simulations that can be used as a qualitative confirmation or collaboration of our CESM1

results. Nevertheless, following the reviewer’s suggestion, we tried to summarize and compare the results from the CESM1 and CMIP models in terms of the impact of sea ice-air coupling on the AMOC, as shown in Fig. R3 (new Fig. 5c) below. The future weakening of the multidecadal variability of LSSDS SIC and AMOC (and other factors) in both the CESM1 1%CO₂ run and CMIP5/6 simulations is broadly similar to that from the CESM1 CTL_FixedIce-minus-CTL differences. Although multidecadal E–P and SSS variations increase in the 23rd century, the LHF-related SSS (and thus AMOC) anomalies become much weaker (new ED Fig. 10j). This further suggests that without the SIC-induced large LHF variations through the sea ice-air coupling, the multidecadal AMOC’s variability would become much weaker in the future.

Fig. R3. (new Fig. 5c) Percentage changes of the SIC, E–P, LHF, SSS, RHO and MLD averaged over the LSSDS region and the AMOC index from the CESM1 CTL_FixedIce minus CTL difference averaged over years 11–490 (red bar, relative to CTL), years 151–224 minus years 11–150 from the CESM1 1%CO₂ run (yellow bars, relative to years 11–150), and 2190–2289 minus 1920–2019 averaged over seven CMIP5 and CMIP6 models (blue bars, relative to 1920–2019; the whiskers denote the inter-model spread).

Dai, A., 2022: Arctic amplification is the main cause of the Atlantic meridional overturning circulation weakening under large CO₂ increases. *Climate Dynamics*, doi:10.1007/s00382-021-06096-x.

3. While the text is reasonably concise, I feel there are too many figures for the audience of a typical nature style article. The present manuscript includes 10 Figures in the main text, 10 Figures in the Extended Data, and 10 Figures in the supplementary information. I would recommend the paper to a more technical journal in climate sciences if there is too much information in the paper that cannot be synthesized for a general audience.

Response: We agree that there are a large number of figures from analyses of many different model simulations in this manuscript. We were hoping that the analyses using different model

simulations would strengthen our key arguments on the important role of sea ice-air coupling in amplifying multidecadal variability in the Arctic and North Atlantic.

We chose this journal because it allows more figures and longer text than other Nature journals to report major new discoveries. We hope that the paper will attract more attention by appearing in *Nature Communications* than in *J. Climate* or *Climate Dynamics* (the two journals we usually publish our research work), so that more colleagues will pay attention to sea ice-air coupling in our climate system. Furthermore, papers in *J. Climate* or *Climate Dynamics* seldom have more than 20 figures and often without supplementary materials, while *Nature* journals allow supplementary figures and tables. This flexibility allows us to present more information to support our arguments.

We have combined some relevant figures to facilitate comparison. We now have 8 figures in the main text, 10 figures in the Extended Data, and 8 figures in the supplementary information.

Other minor comments:

L24-25: can you include estimates of the reduction in variability for all the variables here?

Response: As described in the 2nd paragraph of the Result section (lines 99–104), the variability reduction in the CTL_FixedIce run is around 36–49% for Tas over the LSDS and BKS, ~31% for AMV, ~16% for SIC in the BKS and ~5% for SIC in LSDS. The reduction is even larger in the 1%CO₂_FixedIce run. The AMOC weakens by 20% (line 194) in the CTL_FixedIce run. Given the relatively small reduction in SIC, removed SIC from those two lines (but note that the internal SIC variations in FixedIce runs were not seen by the atmosphere and oceans, which only saw prescribed and fixed SIC) and included some numbers as follows:

“When sea ice is fixed in flux calculations, multidecadal variations are reduced substantially (by 20–50%) not only in Arctic Tas, but also in North Atlantic SST and AMOC.”

Fig. 2: the caption says that only negative values are plotted in c and f, but it is likely misleading to only plot the regions of decrease in SD, not the regions of increase in SD. I think it should be left to readers to see/decide whether the region of increase is unimportant or not. The same comment applies to the stream functions in Fig. 4c and Fig. 8c.

Response: Thanks for pointing this out. We have revised Fig. 2c,f, Fig. 4c, and Fig. 8c (new Fig. 4f) to better show the SD differences. We also similarly revised other relevant figures in Extended data (e.g., new ED Figs. 1, 4, and 10) and supplementary information (e.g., new Figs. 6 and 8).

Method to fix SIC: in my understanding, the paper uses a method to override the water and ice fractions within a grid-box in the coupler module of a coupled ocean-atmosphere model. While this is certainly a new method to decouple the sea ice-air interaction, I am not sure about the claim that the method does not alter any physical laws (L404-405) --- shouldn't this alter the amount of ice or water used in the flux calculation? The authors further explain that this is in analogy to the change of land surface type (L410-415). While this is explained in the method section, I think brief explanations in the main text would help readers understand the physical meaning of fixing sea ice in flux calculation.

Response: The reviewer's understanding is correct that we read in prescribed SIC to override the internal SIC in the coupler module of the CESM1 for estimating the water and ice fractions in calculating the grid-box mean fluxes. Clearly, this is an artificial intervention of the natural climate system. Using the analogy for land surface type change, one may replace the Amazon rainforest with grassland to study the effect of the deforestation in the Amazon. Clearly, such artificially changed land surface type is unrealistic for today's climate, but it does not violate or alter any physical laws in the model, although the grassland would not exist naturally in today's climate in the Amazon (just like our prescribed sea ice cover may not be consistent with the climate condition inside the model). However, we have already included many human-induced changes (such as urban areas and croplands) that would not exist without human intervention in the standard CESM1. Thus, we probably should not consider such an artificial change of the land surface type (or ocean surface type in our case) as an alteration of physical laws in the model. From this perspective, we think our use of fixed SIC does not violate any physical law; rather, it should be viewed as a human-induced change in ocean surface type (similar to prescribing SSTs in Atlantic pacemaker simulations by CESM1) for studying the impact of such a change on the climate. In summary, prescribing a land or ocean surface type (such as a change in vegetation cover, SST or SIC) in a model, albeit unnatural, does not violate any physical laws within the model.

In response to this comment, we made following revisions:

Around lines 92–93: "... , which only allow the atmosphere and oceans to affect sea ice but not the other way **under a human-induced ocean surface type change in analogy to a land surface type change** (see Methods), ..."

Around lines 433–436: "**While both changes are unnatural and due to human intervention, such human-induced changes (e.g., urban surfaces, croplands) are already included in the standard CESM1; thus, we view our prescribed SIC as another human intervention of the climate system for studying its impact on the climate.**"

References

Blackport, R., & Screen, J. A. (2020). Weakened evidence for mid-latitude impacts of Arctic warming. *Nature Climate Change*, 10(12), 1065–1066. <https://doi.org/10.1038/s41558-020-00954-y>

Blackport, R., Screen, J. A., van der Wiel, K., & Bintanja, R. (2019). Minimal influence of reduced Arctic sea ice on coincident cold winters in mid-latitudes. *Nature Climate Change*, 9(9), 697–704. <https://doi.org/10.1038/s41558-019-0551-4>

Reviewer #2 (Remarks to the Author):

I find this study very interesting and well-written and the authors' methodology of fixing sea ice via fluxes through the coupler is novel and has previously contributed to the understanding of Arctic Amplification (Dai et al, 2019). Overall, I believe this paper is suitable for publication; however, I am hoping that the authors can clarify a few issues prior to publication. I have three major comments and a few minor comments:

Response: We thank the reviewer for the positive review and constructive comments, which are in black. Our response is in blue. Revisions to the manuscript are in red.

Major Comment 1:

I find the results very interesting, but at the same time I am struggling a bit with the chicken-and-egg nature of the results. Let's start with multi-decadal sea-ice variability in the LSDS region. The authors argue that this can affect AMOC variability and show that this is the case in their FixedIce runs and in the warming runs after sea-ice has declined in this region. The authors also argue that multi-decadal variability in the BKS region is diminished because of the lack of coupling. But, if variability in the AMOC is diminished, could this not be affecting the variability in the BKS region alone, rather than it being because of the lack of coupling? Is it possible for the authors to fix the sea-ice fluxes just in the LSDS region to see to what extent the variability in the BKS region is affected?

Response: We thank the reviewer for raising a good point on whether the weakened variability in the FixedIce run is due to the lack of the sea ice-air coupling or due to the weakened AMOC itself. As shown in Fig. 1 and discussed in the paper (lines 79–86), in ERA5 the AMV-related SST anomalies are only up to 0.2°C , which is much smaller than the surface air temperature (Tas) anomalies of $\sim 2^{\circ}\text{C}$ over the LSDS and $\sim 1.5^{\circ}\text{C}$ over the BKS. This is also true in the CTL run (Fig. 3a). Thus, the thermal advection associated with the AMV (which is associated with the AMOC) may only explain a small (<10%) fraction of the Tas variations over these regions. For SST, Fig. 2a,d show similar SD as for Tas over the ice margin zones in the CTL run, suggesting that the AMV-induced thermal advection also likely plays a relatively small role for the variations in SST (and thus SIC) over the ice margin zones in the fully coupled run.

As stated around lines 212–214, when a constant SIC is used in calculating surface fluxes in the CTL_FixedIce run, the multidecadal variability of the SIC-associated latent heat flux (LHF) decreases by about 74% over the LSDS (compared to CTL; as shown in Fig. 5), thereby weakening SST variations over this region. This large decrease in LHF variability is caused by the prescribed and fixed sea ice cover (SIC, due to sea ice's insulation effect as a lid), not due to the weakened (by $\sim 20\%$) AMOC in the CTL_FixedIce run (Fig. 5). This is because in order for AMOC to affect winter surface fluxes (and thus SST, Tas and SIC), it has to first change SIC (i.e., the size of the lid covering the warm ocean in winter), but in CTL_FixedIce run, the SIC is fixed by design, so a change in AMOC's strength cannot really affect surface fluxes

and thus the decadal variability in Tas (and SST to a less extent) in the FixeIce run. Therefore, the lack of the coupling, not the weakened AMOC, has to be the main cause of the weakened variability (in comparison with the CTL) over the LSDS and other regions in the FixedIce run.

Furthermore, Fig. 3d shows that the SIC and Tas variations over the LSDS lead the AMV slightly (by ~ 1 year) in both ERA5 and CESM1 CTL run. This suggests that the AMV is unlikely a cause of the decadal variations over the LSDS in the CTL run. Note that the AMOC index has a slightly different lag correlation with LSDS SIC in Fig. 6c, but for its impact on sea ice and Tas, it is the AMV-induced thermal (i.e., SST) advection that matters.

For the variations over the BKS, as stated around lines 138–139, the AMV-associated SST anomalies can be advected from the North Atlantic to the BKS in about 6 years by upper-ocean currents in both observations and our CTL run (ED Fig. 2d; also see Fig. R4 below). As explained above, the AMV-induced SST anomalies can only explain a small fraction of the SST and Tas anomalies over the BKS even in ERA5, which has a stronger connection with the AMV than in the CESM1 (ED Fig. 2d), with the rest coming from local sea ice-air interactions. Thus, we fully expect the variations in BKS SIC and Tas to be much larger and close to the CTL run if only SIC over the LSDS is fixed. Furthermore, the BKS has a small role in modulating the AMOC, which is affected mainly by the sea ice-air coupling from the LSDS to the Nordic Seas (where deep water is formed). Because such a simulation takes several months to complete on our local Linux sever and is not really necessary, we did not perform the suggested simulation.

Fig. R4. (a) Spatial distributions of the lag time (color shading, in years) of the 10–90-year band-pass filtered DJF-mean SST anomalies during 1950–2020 with respect to the AMV index from ERA5, together with the 1990–2008 mean DJF upper-10m ocean currents from SODA ocean reanalysis data (vectors, in cm s^{-1}) over the North Atlantic. A lag of 3 years means that the local SST decadal variation lags the AMV index by 3 years. Only lags with peak correlation coefficients higher than 0.5 between the local SST and AMV index are plotted. (b) Same as a, but for the median lag time (color shading, in years) among nine 50-year (similar to the length of filtered ERA5 data, see Fig. 1d) segments from the 500-year CESM1 CTL run (that is, the lag time and its associated correlation are first calculated for each 50-yr segment)

and the 500-year climatological DJF-mean upper-10m ocean currents (vectors, in cm s^{-1}) in CTL.

In response to this comment, we added the following to the paper around lines 300–305:

“Because the AMV-induced SST variations are relatively small compared with the SST and Tas variations over the LSDS, GNS and BKS in observations and the fully coupled CTL run, and a weakened AMOC and AMV cannot really affect winter surface fluxes over these regions due to fixed SIC in CTL_FixedIce, the weakened variability in CTL_FixedIce has to come mainly from the lack of the local sea ice-air coupling rather than the weakened AMOC.”

Major Comment 2:

Here is an alternative view. The authors argue that the AMV affects BKS sea-ice and the two-way coupling is required to translate into multi-decadal variability in BKS SIC and Tas. However, Figure ED 2d suggests that this connection does not really exist in CESM1 - the correlations are quite weak compared to the reanalysis. So, is it possible that multi-decadal variability in BKS SIC and Tas in CESM1 is simply a local coupled ice-atmosphere mode and not connected to the AMV?

Response: Yes, the multidecadal variability in SIC and Tas over the BKS and other ice margin zones results mainly from local sea ice-air coupling, which amplifies their anomalies (e.g., triggered by AMV) via surface fluxes, rather than due to the thermal advection by AMV (as suggested previously), whose effect on local Tas is small, as we argued in the first paragraph of the Result section (lines 74–86). In fact, this is one of the main points we emphasized in this study, see lines 289–295. Besides this, we also showed that the sea ice-air coupling from the Labrador Sea to the Nordic Seas (but not the BKS) can amplify the variability in AMOC and the AMV.

Unlike LSDS SIC and Tas (Fig. R5a), it is true that the AMV's connection to BKS SIC and Tas is weak in our 500-year CTL run compared with that in ERA5 as shown by ED Fig. 2d; however, the stronger connection seen in ERA5 can be largely captured by nine 50-year segments (similar to the length of the filtered ERA5 reanalysis) of the CESM1 CTL run, although with a wide range of lag time for both BKS SIC and Tas (Fig. R5b). This suggests that the connection of BKS SIC and Tas with AMV does exist in CESM1 but might be unstable over time, especially in such a long-term simulation. Similar results are found in other CMIP5/6 piControl runs (Fig. R5c,d). Overall, the averaged weak connection to BKS should not affect our two main arguments: 1) local SIC and Tas multidecadal variability results mainly from the local sea ice-air coupling over the ice margin zones and 2) this coupling (mainly from the LSDS to the Nordic Seas) can also amplify the variability in AMOC and AMV.

In response to this comment, we updated Fig. 3d and ED Fig. 2d and revised the text around lines 289–295 to the following:

“Our results, summarized in Fig. 8, suggest that although the poleward heat transport in the upper ocean from the northern North Atlantic to the GNS and BKS by the AMOC may trigger small multidecadal anomalies in SST and SIC over the subpolar Atlantic and Arctic regions^{10,13,14}, it is the local sea ice-air two-way interaction and the associated surface fluxes in the subpolar Atlantic and Arctic regions (including the LSDS, GNS and BKS), not the AMOC-induced heat transport itself, that are largely responsible for the large Tas and SIC multidecadal variations in these regions.”

Fig. R5. Lead-lag correlation coefficients of the DJF-mean band (10–90 year) filtered AMV with SIC (black) and Tas (red) over the (a) LSDS and (b) BKS regions averaged over nine 50-year segments (close to the length of filtered ERA5 data, see Fig. 1d) from the 500-year CESM1 CTL run (thin curves with the respective colors for individual 50-year segments, and thick curves are their ensemble mean). (c, d) Same as a and b, respectively, but averaged over seven 500-year piControl simulations from seven CMIP5 and CMIP6 models (thin curves for individual model runs and thick for the ensemble mean). The open circles in a and c (b and d) indicate the peak correlations (i.e., negative for SIC and positive for Tas) from lags -7 to $+7$ (0 to $+15$). Note that the ensemble mean curves in a and b are quite similar to the solid curves in Fig. 3d and ED Fig. 2d (based on the 500-year time series from CTL), respectively.

Major Comment 3:

There are two instances in particular where CESM1 seems to behave either differently from reanalysis (Fig. ED 2d) or differently from other GCMs/ESMs (Fig. 6c versus d). These differences affect the interpretation of the results. I realize the CESM1 is a well-documented and reasonably well-performing model, but I am hoping the authors can provide some clarification or reasoning behind why these differences are acceptable. For example, can the authors reproduce Fig. ED 2d for the CMIP5/CMIP6 models as a point of comparison?

Response: As stated in our response to Major Comment 2, the weak connection of BKS SIC and Tas to AMV as shown in ED Fig. 2d does not affect our main conclusions: 1) local (including BKS) SIC and Tas variability results mainly from local sea ice-air coupling, not from AMV, in the sea ice margin zones; and 2) the sea ice-air coupling from the LSDS to the Nordic Seas (but not the BKS) can amplify the multidecadal variability in AMOC and AMV.

For the comparison with the 500-year piControl runs by the CMIP5/6 models, ED Fig. 2d (CESM1 CTL run) is similar to the ensemble mean from CMIP5/6 models (Fig. R5d above) and both showed a weak connection, on average, between AMV and BKS SIC and Tas variations but the connection can be strong in some of the sampled segments or model runs, suggesting that this observed relationship can be reasonably reproduced by our CESM1 CTL run over certain periods and some of the CMIP5/6 piControl runs. In addition, Fig. 6d (CESM1 CTL run) is also very similar to the ensemble mean curves shown in Fig. 6f from the CMIP models. Fig. 6c (CESM1 CTL run) is also broadly similar to Fig. 6e (CMIP piControl runs), except for the black curve (the AMOC vs. LSDS SIC correlation).

As stated around lines 341–346, CMIP models show considerable spread in the correlations of the decadal-multidecadal variations between AMOC and LSDS SIC in their piControl runs among the CMIP models. This suggests substantial uncertainty in current model simulations of the SIC vs. AMOC relationship examined here, and further investigations using other climate models are needed. We slightly modified the text around line 340 to explicitly state that there is a need to examine the SIC vs AMOC relationship in other models, and also added the following to lines 346–350: “We noticed substantial uncertainty in the connection of BKS SIC and Tas variations to the AMV simulated by the CESM1 (ED Fig. 2d) and other climate models (not shown), but this deficiency should not affect our main conclusion on the critical role of sea ice-air coupling for local SIC and Tas variability, AMV and AMOC, which is not significantly affected to BKS sea ice-air coupling.”

Minor Comments:

Abstract: The wording of the first sentence of the abstract is a bit confusing. It reads as though the authors are saying that the rapid warming of the Arctic is due to internal variability. Of course, much of that signal is a forced anthropogenic signal. So, I believe the authors are just referring to a multi-decadal component. I suggest rephrasing to make this clearer.

Response: The Arctic is warming up because of rising GHGs, but the warming rate varies substantially from decades to decades, with the recent period from the late 1990s showing the largest warming rate. This decadal change in Arctic warming rates (i.e., the multidecadal trend) has been attributed to internal multidecadal variability, such as the multidecadal variations induced by the sea ice-air interactions discussed in this study. In our new study (new ref. 6), we showed the sea ice-air interactions contributed to the recent winter warming (cooling) over the BKS (central Eurasia) from 1992–2012, and the temperature anomalies over the two regions are connected via increased Ural blocking circulation.

In response to this comment, we inserted “partly” after “attributed” (line 19) and cited our new study (ref. 6) to further support the claim made in this sentence.

Line 32: There is some literature that relates multi-decadal variability in the Arctic to forcings other than GHGs, i.e. aerosols (Shindell and Faluvegi, 2009; Fyfe et al. 2013). This forced multi-decadal signal is much larger in the Arctic than the global mean. When the authors detrend based on global mean SAT, to what extent do the aerosol effects are Arctic warming remain as part of the multi-decadal signal (this is only an issue for the reanalysis and historical model runs)?

Response: As described in Methods (lines 460–464), we used global-mean temperature from the CMIP multi-model ensemble mean as the forced *signal* (the *x* variable, not the forced *component*) and regressed it against the target variable (the *y* variable) at each grid point, and used the regressed *y* as the forced component and removed the forced component from the original *y* time series. The regression coefficient varies spatially and thus accounts for different regional responses to the same external forcing time series such as that from aerosols (see Dai et al. 2015, ref. 51). Thus, to a large extent, this regressed response should account for the larger Arctic response to aerosols’ effect. Please note that due to nonlinear interactions between GHG and aerosol forcing in the Arctic (Deng et al. 2020), their combined effect on Arctic temperature may not be the same as the linear combination of those seen in individual forcing runs.

In response to this comment, we added the following to Methods (lines 464–465): “The regression coefficient varies spatially and largely accounts for different regional responses to the same forcing⁵¹.”

Deng, J., A. Dai, and H. Xu, 2020: Nonlinear climate responses to increasing CO₂ and anthropogenic aerosols simulated by CESM. *J. Climate*, **33**, 281-301. <https://doi.org/10.1175/JCLI-D-19-0195.1>

Lines 141-153: This paragraph is based on the argument that BKS sea ice is affected by AMV SST anomalies, but Fig. ED 2d does not really show this for CESM1.

Response: Fig. R4b shown above seems to suggest some connections between AMV and the BKS SST. We have double checked our 500-year CTL run and found that their connections seen in ERA5 (which is one particular sample) can be largely captured by nine 50-year segments (close to the length of the filtered ERA5 data) of the CESM1 CTL run, although with a wide range of the lag time for both BKS SIC and Tas (Fig. R5b). Furthermore, other CMIP5/6 models similarly simulated a weak connection of BKS SIC and Tas with AMV in their 500-year piControl runs (Fig. R5c,d). This suggests that the AMV's connection to BKS SIC and Tas seen in ERA5 does exist in CESM1 and other climate models but might be unstable with time, especially in a long-term simulation.

At these lines, we meant to say that the AMV-induced relatively small SST anomalies can be advected into Arctic ice margin zones in a few years, where it can trigger a SIC anomaly which starts the positive feedback loop described in the paragraph that amplifies the initial SST and SIC anomalies and cause large Tas anomalies over the ice margin zones. We did not mean to say that the local SST and Tas anomalies are mainly caused by the AMV-induced SST advection. We revised the text around lines 138–141 to the following:

“The AMV-associated **relatively small** SST anomalies can be advected from the North Atlantic to Arctic ice margin zones in 2–6 years by upper-ocean currents in both observations and our CTL run (ED Fig. 2d), and the sea ice-air coupling allows sea ice to respond to **and amplify** these ocean-induced SST anomalies.”

Lines 225-234: The lag relationships are quite different in CESM1 compared to CMIP5/CMIP6 - can the authors explain? Why should the reader accept that CESM1 is realistic?

Response: We agree that there are substantial spreads in the simulated SIC vs. AMOC relationship among the models, and further analyses of this relationship in other models are needed (as we pointed this out around lines 341–346). However, Fig. 6c-f show that the CESM1-simulated lag correlations (panels c-d) are comparable those for the multi-model ensemble mean of the CMIP-model simulated lag correlations (panels e-f) for all the cases except for the black curve (for AMOC vs. LSSD SIC correlation) in panels c and e. The AMOC vs. E-P, LHF, SSS, MLD, and RHO correlations are all comparable, and the AMOC vs. AMOC correlations (black curve in panels d, f) are very similar, suggesting similar periods of the AMOC's oscillation in the CESM1 and the CMIP models. Thus, overall the lag relationships in the CESM1 are comparable to the CMIP multi-model ensemble mean, despite the substantial inter-model spread.

Given this, we further showed the SIC-LHF relationship simulated by CMIP5/6 models (Fig. R6 below) and found that their relationship is much weaker (with a large inter-model spread) in these models than that in our CESM1 CTL run (which agrees well with that in ERA5), giving rise to the uncertainty in the simulated SIC-AMOC relationship. Thus, we think that the black curve in Fig. 6c for the CESM1 makes more physical sense than the black curve in Fig. 6e for the CMIP models. This is because when a strong AMOC leads to a positive SIC, the excess ice cover should reduce local surface LHF (Fig. R6) and E–P simultaneously, leading to negative AMOC vs. LHF and E–P correlation with a similar lag as seen in the CESM1 (Fig. 6c) and ERA5 (Fig. R6). In contrast, in some of the CMIP models, a strong AMOC seems to first cause negative anomalies in surface LHF and E–P, and then a positive SIC anomaly, which does not make physical sense as AMOC’s influence on surface fluxes should come through changes in SIC, because winter sea ice acts as an insulation lid separating the warm ocean water from the frigid Arctic air.

In response to this comment, we added Fig. R6 to the Supplementary information as Fig. S3, and the following text around lines 227–229: “..., although substantial spread exists among the models (Fig. 6e,f) likely due to the weak SIC-LHF relationship in these models (Supplementary Fig. S3)”.

Fig. R6. Lead-lag correlation coefficients of the DJF-mean band (10–90 year) filtered SIC with LHF averaged over the LSDS from the ERA5 reanalysis during 1950–2019 (black), the CESM1 CTL run (dark blue), and averaged over seven 500-year piControl simulations from seven CMIP5 and CMIP6 models (with the thin gray lines for individual model runs and the light blue line for the ensemble mean). The dots indicate the correlation coefficient is statistically significant at the 5% level based on a resampling technique (see Methods). Note that the LSDS region is shifted northward by 10° latitudes (i.e., 45°–65°W, 65°–80°N) in ERA5 with respect to that defined in CTL and CMIP5/6 piControl runs (i.e., 45°–65°W, 55°–70°N) to better show the current SIC-LHF relationship, because the sea-ice edge (and thus SIC

variations) over LSDS has retreated further northward under the current climate (cf. Fig. 1b) than that under the pre-industrial climate (cf. Supplementary Fig. S1a).

Lines 242-246: wouldn't the enhanced LHF also lead to a local cooling and sea ice growth? In Fig. 6c, when AMOC leads, there is a positive correlation between SIC and AMOC. I am not exactly sure how this fits into the overall interpretation of the results presented in the schematic in Figure 10?

Response: Yes, in theory the enhanced LHF could cool the surface large enough to cause sea ice growth; but in the CESM1, the increased LHF did not lead to sea ice growth presumably because it did not cool the surface large enough (e.g., due to vertical heat mixing, horizontal heat transport or increased downward LW heating) to reach the freezing point. In fact, Fig. R7 below shows that the SST anomalies mostly positive over the LSDS during years with high LHF. In other words, the enhanced evaporative cooling of the surface is overcome by other warming processes (e.g., vertical and horizontal heat transport and increased downward LW radiation), leading to a net warming of the surface during those years with high LHF.

A comparison of the AMV vs. SIC correlation shown in Fig. 3d and the AMOC vs. SIC correlation shown in Fig. 6c suggests that the AMOC lags AMV (i.e., NASST) by 3–5 years, which is indicated by the lower link between “Intensified AMOC” and “Enhanced deep water formation” in Fig. 8. A warm AMV phase should lead to negative SIC in LSDS with little time lag (Fig. 3d), but because of the time lag between AMV and AMOC, the AMOC vs. SIC correlation is actually positive when AMOC leads by 2 years, or negative when AMOC lags by 5 years (Fig. 6c). The key point is that the negative relationship between NASST (i.e., AMV) and LSDS SIC is almost simultaneous, but the response of the AMOC is delayed by 3–5 years.

In response to this comment, we added the following text:

Around lines 251–253: “Note that the enhanced LHF can cool the surface, but the cooling is overcome by other warming processes, leading to mostly positive SST anomalies during years with large LHF in the CESM1 (Supplementary Fig. S6).”

Caption for Fig. 8 (lines 792–793): “leading to a deeper and stronger AMOC in ~3–5 years (i.e., the AMOC lags the SST anomalies and thus AMV by 3–5 years), ...”.

Fig. R7. (new Fig. S7) The DJF-mean SST (contours, in °C) and 10–90-year band-pass filtered SST anomalies (shading, in °C) averaged over the years with high LHF (local maximum higher than +1 SD) over the LSDS region (outlined in **a** and **b**) based on the CESM1 **(a)** CTL and **(b)** CTL_FixedIce runs from years 11 to 490. A nine-point spatial smoothing was applied in all panels.

Reviewers' Comments:

Reviewer #1:

Remarks to the Author:

First, I thank the authors' efforts to create the new Fig. 5c, which takes into account my suggestion for a quantitative comparison among different experiments. Also, figures are combined to produce a better comparison among different experiments. However, I am a little disappointed that the paper does not address the question about the recent decadal variation in observation, which was the motivation of the paper and its impact. The discussion in the reply and revision, in my opinion, makes the causality from this method confusing, as described below.

In the first round of review, I was concerned about the implications of the results presented in the paper for the causality of the Arctic warming trend from 1990 to present, which has been a focus of the debate in the literature. I appreciate the additional analysis in Fig. R1 and the explanations provided in the response letter. The authors also added text to the discussion section of the paper that "the BKS, where rapid warming was observed from 1992–2012 that is attributed partly to the sea ice-induced multidecadal variability and also contributes to the concurring winter cooling over Eurasia via atmospheric cold advection." The authors also admitted in the response letter that the results are "not thoroughly examined."

While I understand that the authors prefer to leave the detailed analysis to a separate study, I am confused by the causality implied by the method with the sea ice fixed in the flux calculation. The authors have made claims of causality based on this method, which seems to be further elaborated in Dai, A., & Deng (2022).

(i) The sea ice loss is, to some extent, causing the Arctic warming and winter cooling over Eurasia.

(ii) The Eurasian cooling is partly attributed to *internal* multidecadal variability caused by the sea ice-air interaction.

(iii) In the response to my question about the Method to fix SIC, the authors explained that "we think our use of fixed SIC does not violate any physical law; rather, it should be viewed as a *human-induced* change in ocean surface type (similar to prescribing SSTs in Atlantic pacemaker simulations by CESM1) for studying the impact of such a change on the climate."

I think these statements are confusing because (iii) indicates the change in sea ice (fixing the flux calculation) is viewed as human-induced change but (ii) refers to the interaction as the internal variability of the climate system. If the change in sea ice is just part of the internal variability in the sea ice-air interaction, I don't think this is a clean way to separate the cause and feedback. The only way that I can think of is through the feedback analysis of individual components via the chain rule. If we only fix a variable in one place but allow it to change in other places, I am afraid that the change seen in the analysis may be due mostly to the inconsistency of a variable in different components of the model. Having said that, I think the results presented in the paper are technically sound and interesting for more technical journals such as Journal of Climate or Climate Dynamics. I am just not sure that the causality implied in the paper with this method is well established for the role of sea ice in a changing climate.

More on the method to fix SIC:

The authors replied to my comments on the violation of physical laws with an analogy to land surface type change as below.

"Using the analogy for land surface type change, one may replace the Amazon rainforest with grassland to study the effect of the deforestation in the Amazon. Clearly, such artificially changed land surface type is unrealistic for today's climate, but it does not violate or alter any physical laws in the model, although the grassland would not exist naturally in today's climate in the Amazon (just like our prescribed sea ice cover may not be consistent with the climate condition inside the model). However, we have already included many human-induced changes (such as urban areas and croplands) that would not exist without human intervention in the standard CESM1. Thus, we probably should not

consider such an artificial change of the land surface type (or ocean surface type in our case) as an alteration of physical laws in the model. From this perspective, we think our use of fixed SIC does not violate any physical law; rather, it should be viewed as a human-induced change in ocean surface type (similar to prescribing SSTs in Atlantic pacemaker simulations by CESM1) for studying the impact of such a change on the climate.”

I don't quite follow the authors' argument. First, the authors try to link the changes in surface type to climate change, but the investigation on sea ice-air interaction is about the internal variability of the climate system. I am not sure that the authors are thinking about the forced change or unforced change in the framework. It would be helpful to use a more quantitative framework that can be phrased mathematically by the chain rule as for climate sensitivity. Secondly, I cannot follow the analogy between the land surface type and sea ice. Sea ice acts as the interface between the air-ocean interaction, which the proposed method break down the strong coupling between the atmosphere with active ocean heat transport, but the surface type does not play such an important role in the coupling as the heat diffusion in the land is much weaker and less effective.

Reviewer #2:

Remarks to the Author:

I thank the authors for addressing my comments. Although I am still finding it somewhat challenging to isolate all the different elements that are changing and how they interact (it would be nice to see a follow-up study that attempts to cleanly separate the influence of the different regions), I am satisfied with the replies and I have no further comments.

Response to Reviewers' comments:

Reviewer #1 (Remarks to the Author):

First, I thank the authors' efforts to create the new Fig. 5c, which takes into account my suggestion for a quantitative comparison among different experiments. Also, figures are combined to produce a better comparison among different experiments. However, I am a little disappointed that the paper does not address the question about the recent decadal variation in observation, which was the motivation of the paper and its impact. The discussion in the reply and revision, in my opinion, makes the causality from this method confusing, as described below.

Response: We thank the reviewer for reviewing the revised manuscript and for the further comments, which are concentrated on 1) the lack of detailed analysis of the recent decadal variations in observations, and 2) the interpretation of our FixedIce experiments. To further address the first concern, we have performed additional new analyses and added new text (lines 105–120) and a new figure (ED Fig. 4) to the manuscript and have provided more explanation below. The 2nd concern is addressed below the specific comments. Our further revised text in the manuscript is in blue color, while the red text is from the previous (1st) revision.

In the first round of review, I was concerned about the implications of the results presented in the paper for the causality of the Arctic warming trend from 1990 to present, which has been a focus of the debate in the literature. I appreciate the additional analysis in Fig. R1 and the explanations provided in the response letter. The authors also added text to the discussion section of the paper that “the BKS, where rapid warming was observed from 1992–2012 that is attributed partly to the sea ice-induced multidecadal variability and also contributes to the concurring winter cooling over Eurasia via atmospheric cold advection.” The authors also admitted in the response letter that the results are “not thoroughly examined.”

Response: To address the reviewer's concern on the recent Arctic trend, we have performed new analyses and added new discussions and figures into the main body of the manuscript, as mentioned above.

We agree that the causality of the Arctic warming trend since the 1990s, especially the warming over the BKS and its link to recent winter cooling over Eurasia, has been a major focus of recent research. In Dai and Deng (2022, *Climate Dyn.*), we have examined the role of sea ice-air interactions in causing the recent BKS warming and Eurasian cooling. While the recent warming trends over other Arctic regions besides the BKS also require further

investigation, we think the focus of the on-going debate is mainly on the recent **BKS warming and Eurasian cooling**, rather than the Arctic-wide warming trend or the warming trends over other Arctic regions outside the BKS. Many papers, including Screen and Simmons (2010, *Nature*) and Dai et al. (2019, *Nature Comm.*), have already examined the recent Arctic-wide warming trends since 1979 and concluded that sea-ice loss is a major contributor to the enhanced Arctic warming that occurred mainly over the cold season and mainly over the areas with large sea-ice loss due to enhanced winter heating from the Arctic Ocean after sea ice retreat. Nevertheless, the relative roles of internal variability (such as IPO and AMO) and external forcing in causing recent Arctic (not just BKS) warming trends still require further investigation and we're currently working on that.

While the amplification effect revealed in this manuscript is relevant to the role of internal variability, the focus of this study is on the new discovery of the amplification effect of the sea ice-air interactions on the multidecadal variability in the North Atlantic and Arctic region as implied by its title. We believe this is an important new discovery that deserves to be published in a high-profile journal. We mentioned the recent Arctic warming as one of the motivations to study Arctic climate variability, rather than really to explain it in this manuscript. As noticed by the reviewer, there are already many figures and results in the current version of the manuscript just to reveal this amplification effect; adding more detailed analyses of the recent Arctic-wide warming trends to the manuscript would increase the number of figures more and also dilute our focus on the new discovery.

Nevertheless, following the reviewer's suggestion, we performed additional analyses to discuss the role of internal variability in causing the recent Arctic-wide warming trends. We found that the observed decadal warming trend over the BKS from 1997–2009 ($\sim 2.69^{\circ}\text{C}/\text{decade}$; Fig. R1a) is close to that in our CTL run ($\sim 2.99^{\circ}\text{C}/\text{decade}$; Fig. R1b) over the periods with largest BKS warming (top five percentiles) among all 13-year moving trends. However, when the sea ice-air interactions are cut off in the CTL_FixedIce run, this strongest decadal warming trend is reduced substantially to $1.36^{\circ}\text{C}/\text{decade}$ (by $\sim 55\%$ relative to CTL) (Fig. R1c). We further found that recent decadal warming trend over the whole Arctic or the BKS region is more likely to occur with the sea ice-air coupling than the case without the coupling (Fig. R1d–e). This is because the weakened multidecadal T_{as} variation cannot generate large multidecadal T_{as} trends over the Arctic region (especially BKS) without the sea ice-induced amplification effect through the surface flux changes revealed in this manuscript. However, a thorough analysis on the relative roles of internal variability and external forcing in causing the recent Arctic warming trends, which is not the goal of this study, will require another full-length article and we're working on that in a separate study.

Fig. R1 (new ED Fig. 4) (a–c) Decadal trend maps of 10–90-year band-pass filtered DJF-mean surface air temperature (Tas) anomaly field (in °C per decade) north of 60°N (a) from ERA5 during 1997–2009 (after removing the forced signal, see Methods) and averaged over the 13-year moving periods with the strongest BKS warming trends (top five percentiles) from the CESM1 (b) CTL and (c) CTL_FixedIce runs during years 11–490. The BKS Tas trend seen in (a) ERA5 during 1997–2009 and averaged over the top five percentiles from (b) CTL and (c) CTL_FixedIce runs are also given within the outlined BKS region (as defined in Fig. 1b). The stippling indicates that the trend is statically significant at the 5% level based on a Students’ *t* test in panel a or at least 90% of the selected periods agree on the same sign of trend in panels b–c. (d–e) The probability density function (PDF) of (d) Arctic (north of 65°N) and (e) BKS Tas trends over all 13-year moving periods from the CTL (red solid) and CTL_FixedIce (gray dashed) runs based on a gaussian kernel density estimation. The *x* axis is the 13-year Tas trend (in °C per decade) and the *y* axis is the probability density. The vertical black solid lines (with the corresponding value in the top right) indicate the (d) Arctic and (e) BKS warming trends from ERA5 during 1997–2009, while the vertical red solid and gray dashed lines indicate the 5th percentile values in the CTL and CTL_FixedIce runs, respectively. The occurrence probability for a trend that is similar or larger than the ERA5 trend for the Arctic-mean (BKS) Tas is increased from 2.14% to 5.78% (0.00% to 3.64%) from CTL_FixedIce to CTL. Note that we chose 1997–2009 here because the BKS Tas shows a rapid warming trend over this period (cf. Fig. 1f).

In response to this comment, we moved the last paragraph of Summary and discussion in our 1st revision to the main text around lines 105–121 and revised it to the following, and also added Fig. R1 to the Extended data as new ED Fig. 4:

“The large multidecadal temperature variations induced by the sea ice-air interactions can cause apparent warming or cooling trends over multidecadal periods over many Arctic regions, such as BKS, where a large decadal warming trend was observed from 1997–2009 ($\sim 2.69^{\circ}\text{C}/\text{decade}$; Fig. 1f and ED Fig. 4a). Such a decadal trend is also seen in our CTL run over certain decadal periods of similar length (e.g., the top five percentiles show a mean trend of $\sim 2.99^{\circ}\text{C}/\text{decade}$; ED Fig. 4b). However, these strongest BKS decadal warming trends are reduced substantially (by $\sim 55\%$) to $\sim 1.36^{\circ}\text{C}/\text{decade}$ when the sea ice-air interactions are cut off in CTL_FixedIce (ED Fig. 4c). As a result, the recent decadal warming trend over the Arctic or BKS region seen in ERA5 is much less likely to occur without the sea ice-air coupling than the case with the coupling, with the occurrence probability of a similar or larger trend for the Arctic-mean and BKS trends increased, respectively, from 2.14% to 5.78% and 0.00% to 3.64% from CTL_FixedIce to CTL (ED Fig. 4d–e). This implies that the recent rapid warming in the Arctic or BKS region can arise from the multidecadal variability amplified by the sea ice-air interactions (although it is a small-probability event), which also contributes to the concurring winter cooling over Eurasia via atmospheric cold advection⁶. However, the relative roles of internal variability and external forcing in causing the recent Arctic warming trends still require further investigations.”

While I understand that the authors prefer to leave the detailed analysis to a separate study, I am confused by the causality implied by the method with the sea ice fixed in the flux calculation. The authors have made claims of causality based on this method, which seems to be further elaborated in Dai, A., & Deng (2022).

- (i) The sea ice loss is, to some extent, causing the Arctic warming and winter cooling over Eurasia.
- (ii) The Eurasian cooling is partly attributed to **internal** multidecadal variability caused by the sea ice-air interaction.
- (iii) In the response to my question about the Method to fix SIC, the authors explained that “we think our use of fixed SIC does not violate any physical law; rather, it should be viewed as a

human-induced change in ocean surface type (similar to prescribing SSTs in Atlantic pacemaker simulations by CESM1) for studying the impact of such a change on the climate.”

I think these statements are confusing because (iii) indicates the change in sea ice (fixing the flux calculation) is viewed as human-induced change but (ii) refers to the interaction as the internal variability of the climate system. If the change in sea ice is just part of the internal variability in the sea ice-air interaction, I don't think this is a clean way to separate the cause and feedback. The only way that I can think of is through the feedback analysis of individual components via the chain rule. If we only fix a variable in one place but allow it to change in other places, I am afraid that the change seen in the analysis may be due mostly to the inconsistency of a variable in different components of the model. Having said that, I think the results presented in the paper are technically sound and interesting for more technical journals such as *Journal of Climate* or *Climate Dynamics*. I am just not sure that the causality implied in the paper with this method is well established for the role of sea ice in a changing climate.

Response: We apologize for the confusion. We have revised the relevant text (around lines 92–93 and 443–450) by removing “human-induced” to avoid the confusion. Our change is a modeler-specified perturbation or change in ocean surface type (in the coupler for flux calculations only), rather than a human-induced long-term change in climate forcing.

In Dai and Deng (2022), we removed the externally forced changes using the forced signal (through regression) represented by the multi-model ensemble mean (MMM) of the all-forcing historical simulations by 26 CMIP6 models (see section 2.3.1 of Dai and Deng 2022) from ERA5 data so that we can interpret the residual as caused primarily by internal variability. We then compared the trends from 1992–2012 in the residual ERA5 data with the trends over all 21-year segments in our standard (CTL) and fixedIce control (FixedIce_CTL) runs by the CESM1, and found that only the CTL run (with the sea ice-air interactions) can capture the overall trend patterns over the BKS and Eurasia seen in the residual ERA5 data. From this, we concluded that the recent Eurasian winter cooling was partly caused by the internal multidecadal variability amplified by Arctic sea ice-air interactions. Thus, the FixedIce_CTL run was not used to determine whether the recent change is due to natural variability or due to external/human forcing.

In Dai and Deng (2022) and in this manuscript, the FixedIce_CTL run is used as a case where the sea ice-air two-way interactions were artificially cut off, so that the difference between the CTL and FixedIce_CTL runs can be attributed to the sea ice-air interactions existed only in the CTL run, as all other processes were not altered by us in these runs. This is the only purpose of the FixedIce_CTL run in these studies. Our use of “human-induced”

change may be a bit confusing (and led the reviewer to link it to natural vs. human-induced changes), but it really refers to artificially induced changes, like in all numerical experiments in which the modeler changes something artificially in order to study its impact by comparing it to the normal case without such a change. This is a standard modeling approach. The trick is how to implement such a change, which might be inconsistent with the internal model physics.

For example, one can simply reset or relax model internal sea ice cover to a fixed value (with a seasonal cycle) at each time step inside CESM1, but the internal model physics may imply a different ice cover at that time step, leading to some artificial melting or growth of the sea ice that would disrupt the conservation of water and energy in the model. On the other hand, if one artificially replaces the Amazon rainforest with grassland inside CESM1, this won't cause any problems because the CESM1 does not calculate the vegetation type dynamically and it relies on the user to provide the land type over the Amazon, even though the current climate may not allow grassland to exist in the Amazon if the model were to calculate its own vegetation type using a dynamic vegetation model (which is not case for CESM1). Arctic sea ice is dynamically calculated inside the sea ice model of CESM1; thus, one should avoid to change the internal sea ice cover because that will cause inconsistency between the specified and internally-calculated sea ice, which would lead to artificial sea ice melting or growth. In our FixedIce runs, we did not alter the internal ice cover (so that there would be no artificial ice melting or growth); instead, we only fixed the ice cover in the *coupler* of CESM1 for determining the ice and water fractions for use as the weights for calculating the gridbox-mean fluxes. Thus, "FixedIce" actually means fixing the ice cover in the *coupler* for flux calculations (which does not alter the conservation of mass and energy), rather than fixing it in the ice model or other places of the CESM1. As stated in the manuscript (line 430), internal sea ice is still allowed to evolve dynamically in our FixedIce runs.

We made the analog to a land cover type change because they both represent an effect on surface fluxes due to a change in surface type. Clearly, they have differences; e.g., the internal ice cover was not changed by us and was allowed to vary over time in our FixedIce runs, while the specified deforestation becomes an internal part of the model and will not change without a dynamic vegetation model (which is the case for CESM1). We have revised the text (lines 443–450) to point out these points.

More on the method to fix SIC:

The authors replied to my comments on the violation of physical laws with an analogy to land surface type change as below.

“Using the analogy for land surface type change, one may replace the Amazon rainforest with grassland to study the effect of the deforestation in the Amazon. Clearly, such artificially changed land surface type is unrealistic for today’s climate, but it does not violate or alter any physical laws in the model, although the grassland would not exist naturally in today’s climate in the Amazon (just like our prescribed sea ice cover may not be consistent with the climate condition inside the model). However, we have already included many human-induced changes (such as urban areas and croplands) that would not exist without human intervention in the standard CESM1. Thus, we probably should not consider such an artificial change of the land surface type (or ocean surface type in our case) as an alteration of physical laws in the model. From this perspective, we think our use of fixed SIC does not violate any physical law; rather, it should be viewed as a human-induced change in ocean surface type (similar to prescribing SSTs in Atlantic pacemaker simulations by CESM1) for studying the impact of such a change on the climate.”

I don’t quite follow the authors' argument. First, the authors try to link the changes in surface type to climate change, but the investigation on sea ice-air interaction is about the internal variability of the climate system. I am not sure that the authors are thinking about the forced change or unforced change in the framework. It would be helpful to use a more quantitative framework that can be phrased mathematically by the chain rule as for climate sensitivity.

Response: As explained above, our use of the CTL and FixedIce_CTL experiments is just like other numerical sensitivity experiments in which the modeler changes something artificially in order to study its impact by comparing it with the case without this change. This is a very standard modeling approach. Again, we did not introduce any forced change in the framework as the ice cover was not fixed but allowed to change in the ice model in our FixedIce runs, and our artificial change is the use of a fixed ice cover in flux calculations in the *coupler*, so that variations and long-term changes in ice cover won’t affect surface fluxes. In other words, the only intervention is the surface flux exchange in the model *coupler*, and the purpose of this intervention is to reveal the impact of sea ice variations seen in CTL by looking at the CTL minus FixedIce_CTL difference. This also effectively cuts off the two-way sea ice-air interaction, so that we can quantify its impact by comparing to the CTL run. The use of the analog to a land cover change meant to say that they both represent an effect from a surface type change on surface fluxes. To avoid its implication for human influence and natural variability, we have revised the text (lines 446–450) to avoid this confusion.

Secondly, I cannot follow the analogy between the land surface type and sea ice. Sea ice acts as the interface between the air-ocean interaction, which the proposed method break down the strong coupling between the atmosphere with active ocean heat transport, but the surface type does not play such an important role in the coupling as the heat diffusion in the land is much weaker and less effective.

Response: We agree that the underlying land and ocean will induce different interactions between our FixedIce run and a deforestation run. The main point we tried to make is that they both represent an artificial change in surface type that mainly affects surface fluxes and the interactions with the atmosphere. We have revised the relevant text around lines 443–444.

Reviewer #2 (Remarks to the Author):

I thank the authors for addressing my comments. Although I am still finding it somewhat challenging to isolate all the different elements that are changing and how they interact (it would be nice to see a follow-up study that attempts to cleanly separate the influence of the different regions), I am satisfied with the replies and I have no further comments.

Response: We greatly appreciate the constructive comments from the reviewer.

Reviewers' Comments:

Reviewer #1:

Remarks to the Author:

This is my third time to review the manuscript. First, I thank the authors' efforts to address my previous comments about the interpretation of the FixedIce experiment. I wanted to reiterate that the results presented in the paper are technically sound and that they will contribute to our understanding of the role of Arctic Sea ice in the climate system. But I am not sure that the causality implied in the paper with this method is well established for the role of sea ice in a changing climate, especially for the debate on the recent Arctic variability and the connection to midlatitudes.

In the reply, the authors stated that

"Many papers, including Screen and Simmons (2010, Nature) and Dai et al. (2019, Nature Comm.), have already examined the recent Arctic-wide warming trends since 1979 and concluded that sea-ice loss is a major contributor to the enhanced Arctic warming that occurred mainly over the cold season and mainly over the areas with large sea-ice loss due to enhanced winter heating from the Arctic Ocean after sea ice retreat."

I think that an alternative view (Blackport & Screen, 2019, 2020) has suggested that the sea ice change is partly caused by changes in the atmosphere rather than atmospheric variability forced by the changes in sea ice. While the debate is not resolved, fixing the sea ice only in the coupler of a climate model may not be sufficient to identify the causality between the two, as the sea ice variability itself could be the result of air-ocean interaction rather than the cause. While the manuscript is technically sound and interesting, my overall impression is that the chains of causality and feedback described in the paper are rather complicated, as compared with the standard feedback analysis of polar temperature.

References

Blackport, R., & Screen, J. A. (2020). Weakened evidence for mid-latitude impacts of Arctic warming. *Nature Climate Change*, 10(12), 1065–1066. <https://doi.org/10.1038/s41558-020-00954-y>
Blackport, R., Screen, J. A., van der Wiel, K., & Bintanja, R. (2019). Minimal influence of reduced Arctic sea ice on coincident cold winters in mid-latitudes. *Nature Climate Change*, 9(9), 697–704. <https://doi.org/10.1038/s41558-019-0551-4>

Reviewer #1 (Remarks to the Author):

This is my third time to review the manuscript. First, I thank the authors' efforts to address my previous comments about the interpretation of the FixedIce experiment. I wanted to reiterate that the results presented in the paper are technically sound and that they will contribute to our understanding of the role of Arctic Sea ice in the climate system. But I am not sure that the causality implied in the paper with this method is well established for the role of sea ice in a changing climate, especially for the debate on the recent Arctic variability and the connection to midlatitudes.

In the reply, the authors stated that

“Many papers, including Screen and Simmons (2010, Nature) and Dai et al. (2019, Nature Comm.), have already examined the recent Arctic-wide warming trends since 1979 and concluded that sea-ice loss is a major contributor to the enhanced Arctic warming that occurred mainly over the cold season and mainly over the areas with large sea-ice loss due to enhanced winter heating from the Arctic Ocean after sea ice retreat.”

I think that an alternative view (Blackport & Screen, 2019, 2020) has suggested that the sea ice change is partly caused by changes in the atmosphere rather than atmospheric variability forced by the changes in sea ice. While the debate is not resolved, fixing the sea ice only in the coupler of a climate model may not be sufficient to identify the causality between the two, as the sea ice variability itself could be the result of air-ocean interaction rather than the cause. While the manuscript is technically sound and interesting, my overall impression is that the chains of causality and feedback described in the paper are rather complicated, as compared with the standard feedback analysis of polar temperature.

Response: We agree that the sea ice variability can partly result from the sea ice/ocean-air interactions, which has been emphasized in the manuscript. In our FixedIce_CTL run, the atmosphere and oceans see a fixed ice cover (with a mean seasonal cycle), so that the two-way sea ice/ocean-air interactions are cut off, which resulted in reduced variability in sea ice, air temperature, NASST and AMOC. From this, we concluded that the two-way interactions can amplify the multidecadal variations in sea ice, surface air temperature, NASST, and AMOC. Please note that we did not say that the sea ice is the cause of these variations (although we did say that the sea-ice cover change is a major cause of surface flux variations as it acts as a lid); rather, we said that the sea ice/ocean-air two-way interactions can amplify these variations, like the air-sea interactions for amplifying ENSO-related SST anomalies in the tropical Pacific. We explicitly describe the feedback loop through which the two-way interactions can amplify the variations in sea ice cover, air temperature, NASST and AMOC in the manuscript. In this feedback loop, the lower troposphere can contribute to the variability in sea ice and the other fields through changes in downward LW radiation, which is consistent with the view of atmospheric influences on sea ice suggested by Blackport & Screen (2019, 2020) and others.

Regarding the long-term changes in Arctic sea ice cover (SIC) and air temperature, which are not the focus of this manuscript, in Dai et al. (2019) we concluded that sea ice melting is necessary for the existence of large Arctic amplification (AA) under increasing CO₂, based on several lines of evidence, including (but not limited to) the AA differences between a CESM1 standard 1%CO₂/yr run with fully coupled sea ice and another similar run with fixed sea ice cover in calculating the surface fluxes. In the FixedIce 1%CO₂ run, the use of fixed ice cover is a way to minimize the impact of sea-ice loss on surface fluxes under rising CO₂, which yielded little AA. This suggests that surface flux changes induced by sea-ice loss is critical for the creation of large AA, in addition to evidence based on analyses of AA's spatial and seasonal patterns, and the temporal and inter-model relationships between SIC and AA across CMIP5 models, and the extended projections from CMIP5 models. We agree that the cause and impacts of AA are still under debate, but they are not the focus of this study as we have tried to explain to the reviewer. For long-term Arctic sea ice loss, we know it is caused by GHG-induced warming, but what processes can contribute to or enhance Arctic warming and contribute to Arctic sea loss is still under debate.

In response to the comment, we added the following text around lines 189-192 (note ref. 24 is Blackport et al. 2019):

“We emphasize that in the positive feedback loop, the lower tropospheric temperature and humidity will be altered by the surface upward fluxes, leading to changes in downward LW radiation, thus providing an atmospheric influence on sea ice and surface temperature²⁴.”

References

- Blackport, R., & Screen, J. A. (2020). Weakened evidence for mid-latitude impacts of Arctic warming. *Nature Climate Change*, 10(12), 1065–1066. <https://doi.org/10.1038/s41558-020-00954-y>
- Blackport, R., Screen, J. A., van der Wiel, K., & Bintanja, R. (2019). Minimal influence of reduced Arctic sea ice on coincident cold winters in mid-latitudes. *Nature Climate Change*, 9(9), 697–704. <https://doi.org/10.1038/s41558-019-0551-4>